# Enhanced SREBP2-driven cholesterol biosynthesis by PKCλ/ι deficiency in intestinal epithelial cells promotes aggressive serrated tumorigenesis

Yu Muta [1,2,3], Juan F. Linares[1], Anxo Martinez-Ordoñez[1], Angeles Duran[1], Tania Cid-Diaz[1], Hiroto Kinoshita [1], Xiao Zhang[1], Qixiu Han[1], Yuki Nakanishi [2], Naoko Nakanishi[4], Thekla Cordes [5,6], Gurpreet K. Arora[7], Marc Ruiz-Martinez [1], Miguel Reina-Campos [8], Hiroaki Kasashima [9], Masakazu Yashiro[9], Kiyoshi Maeda[9], Ana Albaladejo-Gonzalez[10,11], Daniel Torres-Moreno[10,12], José García-Solano[10,11], Pablo Conesa-Zamora [10,12], Giorgio Inghirami[1], Christian M. Metallo [5], Timothy F. Osborne [13], Maria T. Diaz-Meco [1] ✉ & Jorge Moscat [1] ✉

The metabolic and signaling pathways regulating aggressive mesenchymal colorectal cancer (CRC) initiation and progression through the serrated route are largely unknown. Although relatively well characterized as BRAF mutant cancers, their poor response to current targeted therapy, difficult preneoplastic detection, and challenging endoscopic resection make the identification of their metabolic requirements a priority. Here, we demonstrate that the phosphorylation of SCAP by the atypical PKC (aPKC), PKCλ/ι promotes its degradation and inhibits the processing and activation of SREBP2, the master regulator of cholesterol biosynthesis. We show that the upregulation of SREBP2 and cholesterol by reduced aPKC levels is essential for controlling metaplasia and generating the most aggressive cell subpopulation in serrated tumors in mice and humans. Since these alterations are also detected prior to neoplastic transformation, together with the sensitivity of these tumors to cholesterol metabolism inhibitors, our data indicate that targeting cholesterol biosynthesis is a potential mechanism for serrated chemoprevention.

Colorectal cancer (CRC) is the third most common malignancy worldwide[1]. Despite advances in prevention, early detection, and systemic treatment strategies, the prognosis remains poor in the advanced stage[1]. CRC is a complex and heterogeneous disease that can be stratified into different subsets based on anatomical, histopathological, genomic, and transcriptomic features, but it is in need of a better understanding at the mechanistic signaling and metabolic level[2]. CRC originates through two alternative histologically

identifiable premalignant states: conventional adenomas (CA) and the serrated route[3–6]. CAs were initially considered the only mechanism leading to CRC, which features APC inactivation and the association with mutations in other key genes, including p53, KRAS, and members of the TGFβ pathway[7]. However, it is evident now that the serrated pathway is an alternative route to CRC, mostly originating from sessile serrated lesions (SSL)[4–6]. The clinical implications of this type of CRC tumor are enormous because their flat shape poses serious challenges

for their detection and complete endoscopic resection and are considered the main cause of post-colonoscopy/interval cancers[6]. Furthermore, their unclear morphology and diagnostic challenges in pathology lead to a lack of reliable longitudinal observational data on the natural history of SSLs. Therefore, while much is known about the CA-originating CRCs, our understanding of the molecular mechanisms regulating the initiation and eventual progression from serrated lesions is quite limited. In this regard, although several mouse models recapitulate at least partially different aspects of the CA-CRC sequence, only recently serrated CRC has been modeled in mice by the expression of key drivers like KRAS, BRAF, or NOTCH[8–10]. These models have provided invaluable information on the mechanisms of serrated carcinogenesis post-appearance of genetic mutations. However, the cellular and molecular understanding of the chain of events triggered by the spontaneous initiation mechanisms before oncogenic alterations emerged was lacking. In addition, despite reports on the association between alterations in lipid metabolism and serrated lesions[11–13], the extent to which the metabolic changes contribute to serrated tumorigenesis remains to be elucidated.

We have recently developed a mouse model of SSL that rapidly progresses to cancer without the initial ectopic expression of any oncogene or the inactivation of tumor suppressors. This model was based on our observation that the loss of both atypical protein kinase Cs (aPKCs; PKCλ/ι and PKCζ) in the mouse intestinal epithelium led to the spontaneous development of SSL[14]. These tumors showed, like in humans, a preference for the proximal location in the colon and a significantly large proportion progressed to intramucosal and even invasive adenocarcinoma, which contained areas of poorly differentiated or signet-ring cell carcinomas with severe desmoplastic change indicating the highly aggressive nature of those tumors, which were also immune-excluded[14]. The desmoplastic response in these aPKC-deficient serrated tumors was characterized by a profound remodeling of the fibroblast compartment with an accumulation of collagen and hyaluronan, recapitulating the most salient features of human mesenchymal (m)CRC[14,15].

Therefore, this is a unique model system to study the initiation and progression of serrated CRC in an oncogenic-agnostic manner without the confounding predetermined expression of a particular oncogene or other type of stimuli. The human relevance of the role of the aPKCs as repressors of serrated tumorigenesis was further demonstrated by interrogating publicly available bulk and single-cell gene expression datasets, as well as by multiplex immunofluorescence analyzes of large collections of intestinal tumor specimens in our laboratory. Those analyzes established aPKC deficiency as a driver of serrated mCRC[4,14,16]. These studies also revealed a previously unrecognized bottom-up mechanism of transformation whereby the intestinal crypt differentiation hierarchy, although preserved during serrated initiation and progression, is largely "metaplastically" subverted, resulting in the accumulation of a cancer cell population with metaplastic and fetal features (termed tumor fetal metaplastic cell; TFMC)[15]. Importantly, the gene signature of the TFMCs, and key markers expressed by that mouse cell population, allowed us to define transcriptional and immunofluorescence biomarkers that predicted the poorest survival in CRC[15]. These intrinsic features of the aPKC-deficient tumors together with the extensive stromal reaction model the most aggressive CRCs termed CMS4, or more recently iCMS3F[15,17–19].

However, what remained to be determined is the signaling and metabolic pathways activated by aPKC deficiency that trigger the initial steps in the transformation sequence of serrated tumors and that ultimately sustain the aggressive nature of this type of cancer, especially involving the TFMC population. In this regard, a relatively unexplored area of research in CRC is metabolism, which could provide a rich source of new therapeutics targeting phenotypic dependencies created by non-oncogenic vulnerabilities. In this context,

cholesterol metabolism is particularly vital because it not only provides energy sources and essential structural molecules for cell membranes but also modulates various signal transduction pathways, thereby mediating cell survival, tumor growth, and drug resistance[20–22]. Furthermore, cumulative evidence suggests that reversal of a dysregulated lipid metabolism has synergistic tumor-suppressive effects, offering potential therapeutic opportunities for cancers refractory to conventional treatments[23,24]. Several preclinical studies have particularly supported the association between cholesterol metabolism and CRC[25,26]. Cholesterol supplementation in the diet has been shown to promote tumor development in colitis-associated and APC-inactivated models[25]. Also, the knock-down of SREBP2, a master regulator of cholesterol biosynthesis, has been reported to inhibit xenograft tumor growth of human CRC cells[27]. Conversely, overexpression of SREBP2 in the intestine epithelium accelerated tumor growth in APC-mutated tumor models[26].

However, despite all these reports suggesting a central role of cholesterol metabolism in CRC initiation and progression, data are still inconclusive and fragmentary at the epidemiological level. Furthermore, no studies have yet clearly revealed the importance of cholesterol metabolism in CRC, with a focus on its precursor lesions (i.e. CA or SSL). Thus, although it has been shown that obesity was associated with an increased risk of CRC[28,29], and bariatric surgery in patients with obesity reduced the risk of developing CRC[30], surprisingly the effect of statins on CRC mortality or incidence is not totally clear[31]. For instance, several observational studies suggested a reduction in CRC mortality in statin-treated patients[32–34], whereas no clear association was found when selection and immortal-time bias were adjusted[35]. Similarly, the effect of statin consumption on CRC incidence has been disparate and inconsistent[31]. Therefore, statins are not recommended to prevent or reduce CRC mortality[31]. Notwithstanding this apparently contradictory information, these variable results suggest that the appropriate stratification of the disease subsets as well as a more in-depth understanding of the molecular and cellular mechanisms regulating cholesterol metabolism, will resolve the discrepancies and will provide a better conceptual framework to understand the complex landscape of cholesterol metabolism in CRC development and its therapeutics.

Here, we report that cholesterol biosynthesis is reprogrammed in serrated tumors. We demonstrate that low aPKC expression, a driving characteristic of serrated tumors, upregulates the activity of SREBP2 by suppressing the ubiquitin-mediated degradation cascade of its chaperone protein SCAP. Subsequent dysregulated cholesterol biosynthesis enhances cholesterol production and renders tumor cells addicted to this metabolic pathway, creating a therapeutic vulnerability.

## Results

### Cholesterol biosynthesis is upregulated in aPKC-deficient intestinal tumors

To investigate the signaling and metabolic pathways in serrated tumors, we carried out a gene set enrichment analysis (GSEA) comparing the transcriptomes of *Prkci*[f/f]*Prkcz*[f/f];*Villin-Cre* mouse intestinal tumors and wild-type (WT) intestinal tissue, which revealed that cholesterol homeostasis was the highest-ranked gene signature among the 50 gene sets in the "Hallmarks" compilation (Fig. 1a, b). These tumors also showed enrichment in multiple cholesterol metabolism signatures (Fig. 1c and Supplementary Fig. 1a) with enhanced expression of genes related to cholesterol biosynthesis and uptake (Fig. 1d). Importantly, *Prkci*[f/f]*Prkcz*[f/f];*Villin-Cre* non-tumoral intestinal tissue also showed enrichment in cholesterol homeostasis gene signatures compared to the WT intestine (Supplementary Fig. 1b). Consistent with these observations, intestinal epithelial cells (IECs) isolated from tamoxifen-inducible *Prkci*[f/f]*Prkcz*[f/f];*Villin-Cre*ERT2 mice[15] exhibited similar gene expression patterns upon the simultaneous ablation of both aPKCs, demonstrating that dysregulation of cholesterol metabolism is not

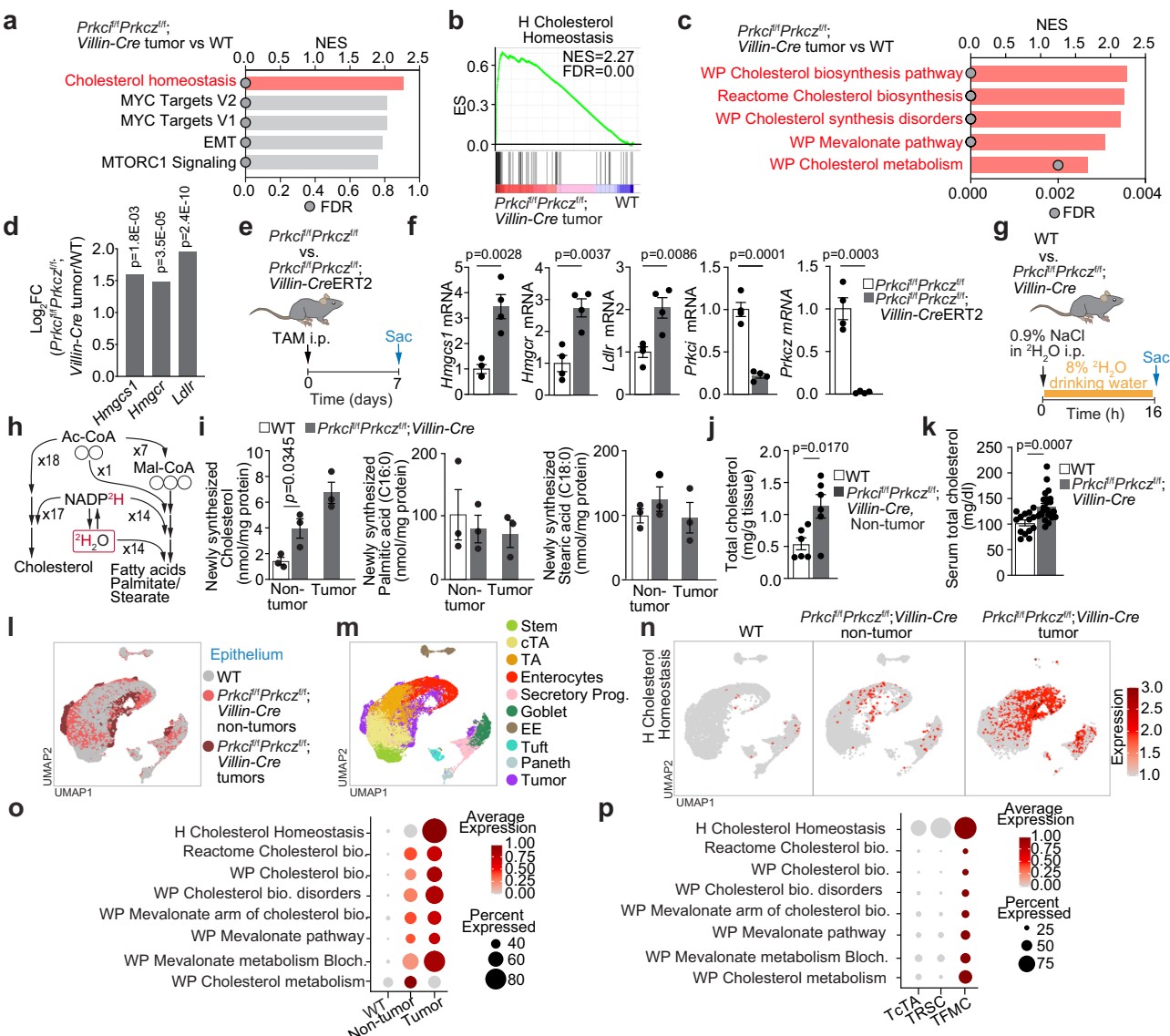

**Fig. 1 | Loss of atypical PKC causes robust reprogramming of cholesterol metabolism by enhancing biosynthesis. a–c** Gene set enrichment analysis (GSEA) results of transcriptomic data from *Prkci^f/f^Prkcz^f/f^;Villin-Cre* mouse small intestinal tumors (*n* = 3 tumors from three distinct mice) versus WT small intestines (*n* = 3 tissues from three distinct mice). The top 5 gene sets in compilation H (MsigDB) (**a**), GSEA plot (**b**), and GSEA results related to cholesterol metabolism (**c**) are shown. **d** RNA-seq data showing the log2-fold changes between *Prkci^f/f^Prkcz^f/f^;Villin-Cre* tumors (*n* = 3 tumors from three distinct mice) and WT tissue (*n* = 3 tissue from three distinct mice). **e, f** Inducible deletion of aPKCs in mouse small intestinal epithelial cells (IECs). *Prkci^f/f^Prkcz^f/f^;Villin-Cre*ERT2 or *Prkci^f/f^Prkcz^f/f^;* mice were injected intraperitoneally with 6 mg of tamoxifen for 7 days. Schematic (**e**) and qPCR (**f**) (*n* = 4 mice per group). **g–i** Metabolic tracing with deuterium water (²H₂O). In vivo experiment schematic (**g**), schematic depicting ²H (red) labeling from deuterated water tracer (²H₂O) and incorporation into newly synthesized cholesterol and fatty acids (**h**), and newly synthesized lipids (**i**) in tumoral (*n* = 3 tumors from three distinct mice) and non-tumoral tissues in the small intestines of *Prkci^f/f^Prkcz^f/f^;Villin-Cre* and WT mice (*n* = 3 tissues from three distinct mice per group). Open circles represent [¹²C]carbon atoms. **j** Total cholesterol content in the small intestine of *Prkci^f/f^Prkcz^f/f^;Villin-Cre* and WT mice (*n* = 6 mice per group). **k** Serum total cholesterol levels in *Prkci^f/f^Prkcz^f/f^;Villin-Cre* (*n* = 23) and WT (*n* = 14) mice. **l–p** scRNA-seq of mouse small intestinal epithelial cells (*n* = 3 tissues from three distinct WT mice, *n* = 1 tissue from one *Prkci^f/f^Prkcz^f/f^;Villin-Cre* mouse, and *n* = 5 tumors from two distinct *Prkci^f/f^Prkcz^f/f^;Villin-Cre* mice). **l, m** Uniform manifold approximation and projection (UMAP) plots of in *Prkci^f/f^Prkcz^f/f^;Villin-Cre* mouse small intestinal epithelial cells colored by tissue origins (**l**) and cell types (**m**). **n** UMAP feature plots of epithelial cells colored by the expression of hallmark cholesterol homeostasis gene set. **o, p** Dot plots of indicated gene signatures against tissue origins (**o**) or tumor cell subtypes in the tumor compartment (**p**). Data were presented as mean ± SEM. Wald test in **d** and two-tailed, unpaired Student's t-test in **f**, **i**, **j**, and **k**. Source data are provided as a Source Data file.

limited to the tumors but can be broadly observed in the non-neoplastic intestine (Fig. 1e, f). This suggests that this pathway might be critical during tumor initiation as an early event upon aPKC loss. We next performed an in vivo metabolic tracing experiment in *Prkci^f/f^Prkcz^f/f^;Villin-Cre* and WT mice to determine the impact of aPKC loss on intestinal lipid biosynthesis using deuterium water (²H₂O) as a tracer (Fig. 1g, h). *Prkci^f/f^Prkcz^f/f^;Villin-Cre* tumors newly synthesized considerably more intestinal cholesterol than WT mice (Fig. 1i).

Notably, de novo cholesterol biosynthesis was also significantly increased in non-tumoral intestines from *Prkci^f/f^Prkcz^f/f^;Villin-Cre* mice (Fig. 1i). In contrast, syntheses of palmitate and stearate were not significantly affected, indicating no difference in fatty acid synthesis (Fig. 1i). In keeping with these observations, cholesterol levels were elevated in *Prkci^f/f^Prkcz^f/f^;Villin-Cre* intestinal tissue (Fig. 1j), as well as in serum (Fig. 1k). Under normal basal conditions, intracellular cholesterol levels and its biosynthesis are tightly controlled[36]. The fact that

cholesterol biosynthesis remained high in the aPKC-deficient intestinal tissue (Fig. 1i), despite the increased cholesterol accumulation (Fig. 1j), suggests the disruption of its homeostatic regulation, which we hypothesize is critical for serrated tumorigenesis.

## Activation of the cholesterol pathway is a critical feature associated with cell metaplasia

We then characterized the aPKC-deficiency-driven dysregulated cholesterol metabolism at the single-cell resolution by integrating our previously reported scRNA-seq data from *Prkci*[f/f]*Prkcz*[f/f];*Villin-Cre* mouse intestinal tumors[15] with non-tumoral *Prkci*[f/f]*Prkcz*[f/f];*Villin-Cre* and WT intestines (Fig. 1l). Epithelial cell re-clustering and mapping of marker gene expression classified cells into tumor and non-tumor cell populations. The non-tumor epithelial cell categories were stem, cycling transient amplifying (cTA), TA, enterocytes, secretory progenitors, goblet, enteroendocrine (EE), tuft, and Paneth (Fig. 1m). Re-clustering of epithelial cells and epithelial pseudo-bulk transcriptomes revealed that the cholesterol signatures were most prominently enriched in *Prkci*[f/f]*Prkcz*[f/f];*Villin-Cre* tumor cells, followed by *Prkci*[f/f]*Prkcz*[f/f];*Villin-Cre* non-tumoral epithelial cells and WT epithelial cells (Fig. 1n–o and Supplementary Fig. 1c), which is in agreement with the enhanced cholesterol biosynthesis observed in the metabolic tracing experiments (Fig. 1g–i). TA, enterocytes, and goblet cells from *Prkci*[f/f]*Prkcz*[f/f];*Villin-Cre* mice displayed a comparably high expression of cholesterol signatures in the non-tumor compartment (Supplementary Fig. 1d). In the tumor compartment, we identified three subpopulations[15], including the previously reported tumor fetal metaplastic cells (TFMC), tumor revival stem cells (TRSC), and tumor cycling TA cells (TcTA) (Supplementary Fig. 1e), where TFMCs showed the highest expression in multiple cholesterol-related gene signatures (Fig. 1p and Supplementary Fig. 1f–i). This observation is of great significance because of all the cancer cell populations in the *Prkci*[f/f]*Prkcz*[f/f];*Villin-Cre* tumors, the TFMC is the cell cluster that includes markers of metaplastic differentiation and the most significantly associated with the poorest prognosis in human CRCs[15]. In line with this observation, mapping of tumor cells showed that cholesterol and metaplastic features overlapped at the single cell level (Supplementary Fig. 1j, k). These results underscore our hypothesis on the crucial role of upregulated cholesterol metabolism in serrated CRCs.

## PKCλ/ι loss is the primary driver for enhanced cholesterol metabolism

Given that the combined loss of both aPKCs resulted in significant dysregulation of the cholesterol metabolism in tumoral and non-tumoral cells, we next dissected the individual roles of each aPKC in this pathway. To this end, we interrogated the bulk transcriptome of mouse IECs from WT, *Prkcz*[f/f];*Villin-Cre*, *Prkci*[f/f];*Villin-Cre*, or *Prkci*[f/f]*Prkcz*[f/f];*Villin-Cre*. GSEA demonstrated that the cholesterol signature was significantly enriched in *Prkci*[f/f];*Villin-Cre* and *Prkci*[f/f]*Prkcz*[f/f];*Villin-Cre* but not in *Prkcz*[f/f];*Villin-Cre* (Supplementary Fig. 2a–f). To investigate the potential cell-autonomous effect of aPKC loss in tumor cholesterol metabolism, we employed mouse tumor organoids (MTO), in which the aPKCs were deleted or not by CRISPR/Cas9[15,37]. RNAseq data from these organoids established that the simultaneous inactivation in MTOs of both aPKCs or that of *Prkci*, but not of *Prkcz*, was sufficient to drive the enrichment in the cholesterol signature and to upregulate the expression of cholesterol biosynthesis genes (Fig. 2a–d and Supplementary Fig. 2g, h). Furthermore, the upregulation of the cholesterol metabolic genes was also demonstrated in aPKC-deficient HCT116 (Fig. 2e) and 293 T cells (Fig. 2f), which was also driven by the single knockout of PKCλ/ι in both cell lines (Fig. 2g, h). Both PKCλ/ι- and aPKC-deficient cells also showed increased intracellular cholesterol levels (Fig. 2i and Supplementary Fig. 2i). Furthermore, the quantification of labeled metabolites using mass spectrometry from cells incubated with [U-$^{13}$C$_6$]

glucose demonstrated that similar to aPKC-deficient intestines in vivo (Fig. 1g-i), PKCλ/ι-deficient cells had upregulated cholesterol biosynthesis as determined by the incorporation of $^{13}$C carbon, with no increase in fatty acid synthesis (Fig. 2j–n). These data demonstrated that the loss of PKCλ/ι faithfully recapitulated the elevated cholesterol biosynthesis observed in aPKC-deficient cells, demonstrating that PKCλ/ι loss is sufficient to act as a primary driver for the metabolic reprogramming of *Prkci*[f/f]*Prkcz*[f/f];*Villin-Cre* tumors.

## aPKC deficiency induces the processing and activation of SREBP2

Next, we sought to identify the precise mechanism whereby aPKC deficiency induces cholesterol metabolic reprogramming. A more detailed inspection of the RNAseq analysis of *Prkci*[f/f]*Prkcz*[f/f];*Villin-Cre* mouse tumors revealed that the genes in the cholesterol pathway exhibiting upregulated expression, all were targets of SREBP2, a master transcription factor of cholesterol biosynthesis[36] (Fig. 3a). Consistently, GSEA validated the enrichment of SREBP2 target signatures in aPKC-deficient tumors (Supplementary Fig. 3a, b). In good agreement with the primary roles of PKCλ/ι on dysregulated cholesterol metabolism (Fig. 2), SREBP2 target gene expression was also upregulated in PKCλ/ι-deficient as in aPKC-deficient IECs, whereas there was no significant change driven by PKCζ-deficiency (Supplementary Fig. 3c). Furthermore, epithelial pseudo-bulk transcriptomes from the aforementioned scRNA-seq data demonstrated the upregulation of these SREBP2 targets in aPKC-deficient non-tumor and tumor cells at the single-cell level (Supplementary Fig. 3d–f). In line with these transcriptomic profiles, genome-wide assay for transposase-accessible chromatin with sequencing (ATAC-seq) of IECs from the four genotypes revealed increased chromatin accessibility for these SREBP target genes (Fig. 3b) and enrichment in the SREBP2 motif (Fig. 3c) in aPKC- and PKCλ/ι- but not in PKCζ-deficient IECs.

SREBP2 activation is tightly governed by subcellular localization and subsequent cleavage[38]. Thus, when intracellular cholesterol levels decrease, precursor forms of SREBP2 translocate from the ER to the Golgi to be cleaved into the mature forms (nSREBP2). Consistent with SREBP2 activation by aPKC deficiency, we found increased levels of the cleaved form of SREBP2 in *Prkci*[f/f]*Prkcz*[f/f];*Villin-Cre* tumors and, to a lesser extent, in the non-tumoral area (Fig. 3d). Immunohistochemistry analysis demonstrated that loss of PKCλ/ι alone or in combination with the that of PKCζ, was sufficient to induce the nuclear accumulation of SREBP2 in IECs in vivo (Fig. 3e). In addition, *Prkci*[f/f];*Villin-Cre* IECs showed increased SREBP2 cleavage but *Prkcz*[f/f];*Villin-Cre* IECs did not (Fig. 3f). We then used intestinal organoids from a tamoxifen-inducible *Prkci*[f/f]*Prkcz*[f/f];*Villin-CreERT2* mouse model[15] (Fig. 3g), in which we demonstrated that the simultaneous deletion of both aPKCs by their induced genetic knock out (KO) resulted in increased levels of mature SREBP2 (Fig. 3h). Additionally, PKCλ/ι-deficient 293 T cells also displayed upregulated SREBP2 processing (Fig. 3i) and enhanced nuclear SREBP2 levels (Fig. 3j, k), which collectively support our hypothesis that PKCλ/ι loss, either alone or in combination with that of PKCζ, is sufficient to induce the cholesterol biosynthesis pathway in several cell systems. Immunofluorescence analyzes demonstrated enhanced nuclear accumulation of SREBP2 in the *Prkci*[f/f]*Prkcz*[f/f];*Villin-Cre* mouse tumor cells in vivo, further supporting the conclusion that aPKC deficiency produces the augmented translocation and activation of SREBP2 under physiologically relevant conditions (Fig. 3l, m). To further explore the specific tumoral cell population with the highest SREBP2 activation in aPKC-deficient tumors, scRNAseq data interrogation showed the upregulation of SREBP2 targets in TFMCs (Fig. 3n and Supplementary Fig. 3g). Furthermore, OPAL multiplex immunofluorescence revealed nuclear SREBP2 expression correlating at the single-cell level with ANXA10, a marker for serrated tumors and TFMCs[15] (Fig. 3o, p).

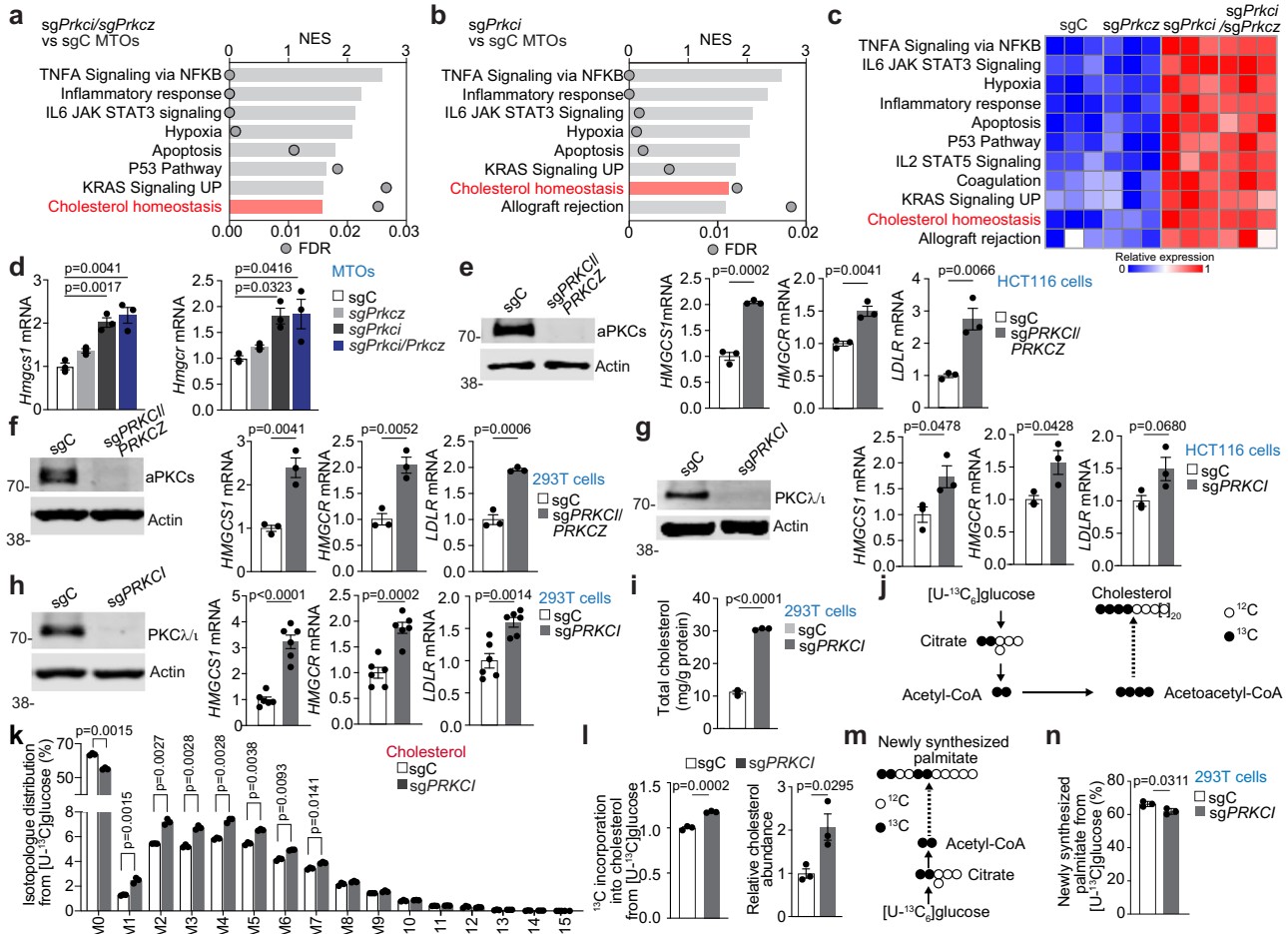

**Fig. 2 | Loss of PKCλ/ι is sufficient for the dysregulated cholesterol metabolism.**
**a**, **b** GSEA results of the top 8 gene sets in compilation H for sg*Prkci*/sg*Prkcz* versus sgC (**a**) and sg*Prkci* versus sgC (**b**) mouse tumor organoids (MTOs) (*n* = 3 biological replicates per group). **c** GSVA results of MTOs. **d** qPCR of MTOs (*n* = 3 biological replicates per group). **e** Immunoblotting and qPCR in sg*PRKCI*/*PRKCZ* and sgC HCT116 cells cultured in 5% lipoprotein depleted serum (LPDS) with 5 μM of lovastatin for 24 h (*n* = 3 biological replicates). **f** Immunoblotting and qPCR in sg*PRKCI*/*PRKCZ* and sgC 293 T cells cultured in 1% LPDS for 24 h (*n* = 3 biological replicates). **g** Immunoblotting and qPCR in sg*PRKCI* and sgC HCT116 cells cultured in 5% LPDS with 5 μM of lovastatin for 24 h (*n* = 3 biological replicates). **h** Immunoblotting and qPCR in sg*PRKCI* and sgC 293 T cells cultured in 1% LPDS for 24 h (*n* = 6 biological replicates). **i** Total cholesterol content in sg*PRKCI* and sgC 293 T cells cultured in 1% LPDS for 40 h (*n* = 3 biological replicates). **j** Schematic

depicting labeling on cholesterol from [U-$^{13}$C$_6$]glucose. Closed black circles represent [$^{13}$C]carbon; open circles represent [$^{12}$C]carbon atoms. **k** Isotopologue distribution of cholesterol from $^{13}$C glucose in sg*PRKCI* and sgC 293 T cells cultured in 1% LPDS for 24 h (*n* = 3 biological replicates). **l** Relative incorporation of $^{13}$C into cholesterol from [U-$^{13}$C] glucose and relative cholesterol abundance in sg*PRKCI* and sgC 293 T cells cultured in 1% LPDS for 24 h (*n* = 3 biological replicates). **m** Schematic depicting labeling on palmitate from [U-$^{13}$C$_6$]glucose. Closed black circles represent [$^{13}$C]carbon; open circles represent [$^{12}$C]carbon atoms. **n** Newly synthesized palmitate from [U-$^{13}$C$_6$]glucose in sg*PRKCI* and sgC 293 T cells cultured in 1% LPDS for 24 h (*n* = 3 biological replicates). Data were presented as mean ± SEM. One-way ANOVA and post hoc Tukey's test (**d**) and Two-tailed, unpaired Student's t-test (**e**–**i**, **k**, **l**, **n**). Source data are provided as a Source Data file.

## Dysregulated cholesterol metabolism is associated with low aPKC expression in human serrated tumors

To determine the human cancer relevance of our findings, we interrogated publicly available human RNAseq data. This analysis revealed that serrated CRCs exhibited higher enrichment in cholesterol signatures than conventional CRCs (Fig. 4a). In line with this finding, premalignant SSLs showed the highest enrichment scores, followed by tubular adenomas and normal tissues (Fig. 4a). In contrast, there was no difference in the enrichment of fatty acid gene signatures in these analyzes (Fig. 4b). These results indicate that enhanced cholesterol metabolism is a characteristic of human serrated tumorigenesis. SSLs mostly originate in the proximal colon[39]. Consistent with previous studies[40,41], proximal CRCs showed worse overall survival than distal in a large-scale CRC dataset from The Cancer Genome Atlas (TCGA) (Fig. 4c). Proximal CRCs also demonstrated enrichment in cholesterol, SSL, and serrated CRC signatures (Fig. 4d). Importantly, aPKC expression was lower in proximal CRCs (Fig. 4e), and patients with low

PKCλ/ι expression levels had significant enrichment in the cholesterol gene expression signature (Fig. 4f, g). In accordance with the observations that TFMCs displayed the highest enrichment in cholesterol signatures in *Prkci*[f/f]*Prkcz*[f/f];*Villin-Cre* mouse tumor subtypes (Supplementary Fig. 1), gene expression data from CRC patients with enriched TFMC also exhibited upregulated cholesterol gene expression, correlating with SSL, and serrated CRC signatures while those corresponding to tubular adenoma signatures were downregulated (Fig. 4h). Also, patient samples corresponding to the recently proposed iCMS classification[19] revealed that iCMS3 patients had upregulated cholesterol signatures compared to iCMS2 patients (Fig. 4i), highlighting the potential pivotal role of cholesterol metabolism in iCMS3 CRCs, which are more often proximal, serrated, and TFMC-like[15,19]. These findings were further supported at the single-cell level in previously published scRNA-seq datasets of human CRCs (Fig. 4j–m and Supplementary Fig. 4a–d). In addition, using human CRC organoids from surgically resected samples, we demonstrate here that, in agreement with mouse

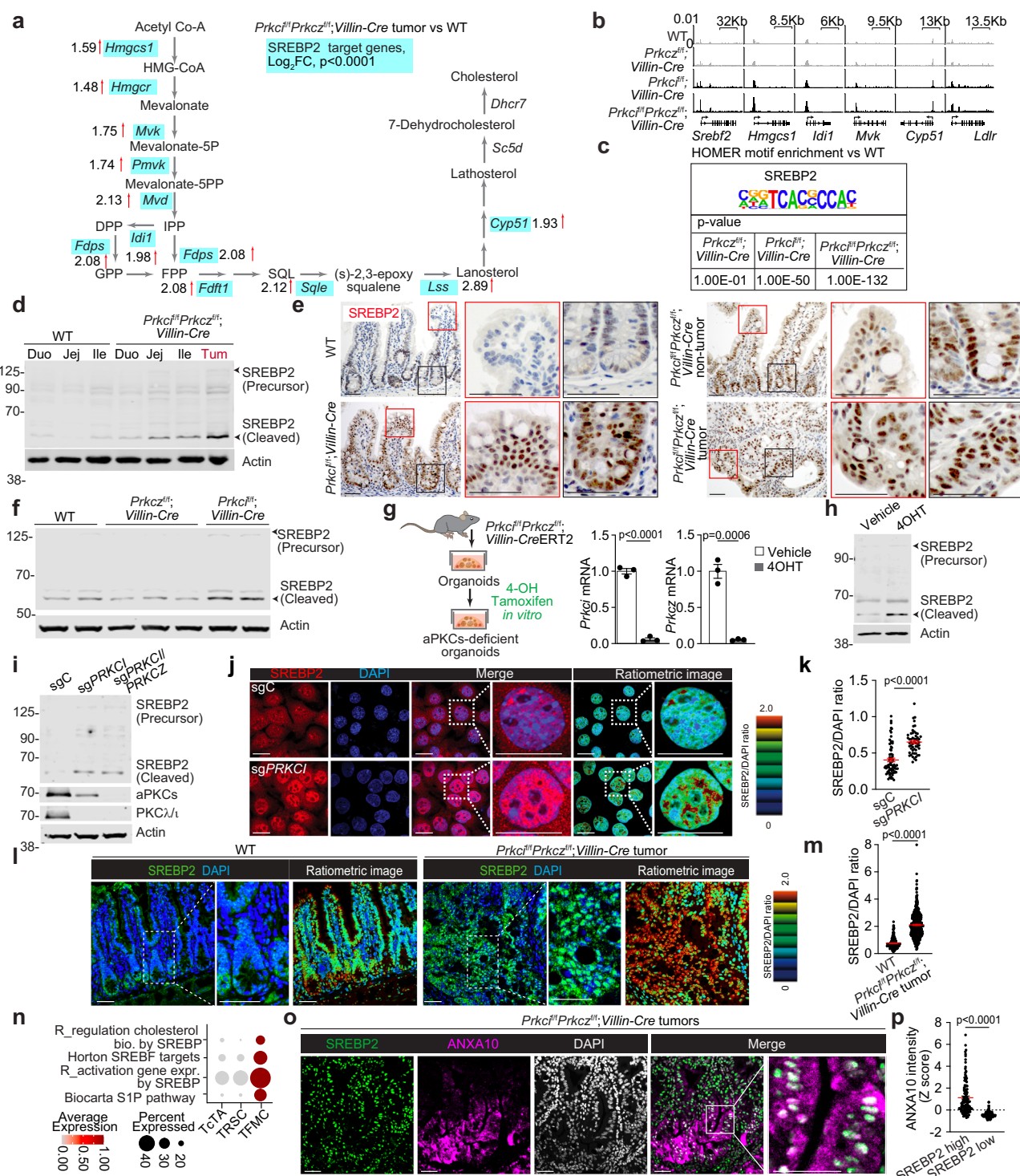

tissues and organoids cultured, human CRC organoids with low aPKC levels exhibited higher expression of SREBP2 target genes concomitant with the metaplastic marker, ANXA10 (Fig. 4n, o).

Finally, we used a tissue microarray (TMA) containing 390 surgically resected CRC tissues to examine the SREBP2 expression at the protein level (Supplementary Fig. 4e). Multiplex staining of this TMA demonstrated that conventional CRCs showed high aPKC expression with minimal ANXA10 and SREBP2 expression, whereas serrated CRCs showed low aPKC expression with concomitant upregulation of ANXA10 and nuclear SREBP2 signals (Fig. 4p). Consistently, multivariate logistic analyzes revealed that high SREBP2 expression was associated with low aPKC expression, serrated histology, and high

ANXA10 expression independent of other histopathological features (Supplementary Fig. 4f–h). Altogether, these results establish the significance of enhanced cholesterol metabolism in poor prognosis human serrated CRC.

## PKCλ/ι-mediated phosphorylation reduces the stability of SCAP

The next series of experiments were designed to uncover the precise molecular mechanisms whereby PKCλ/ι loss induced the processing and activation of SREBP2. Critical to that process is the interaction of SREBP2 with SCAP, a sterol sensing chaperon that translocates with SREBP2 from the ER to the Golgi to induce the processing and activation of SREBP2 through a proteolytic cascade regulated by sterols

**Fig. 3 | SREBP2 processing and translocation is induced by PKCλ/ι deficiency.**
**a** Metabolic map depicting cholesterol biosynthesis pathway showing fold change of SREBP2 target genes from RNAseq data of *Prkci*^f/f^*Prkcz*^f/f^*;Villin-Cre* mouse tumors (*n* = 3 tumors from three distinct mice) versus WT small intestines (*n* = 3 tissue from three distinct mice). **b, c** Genome browser view at promoter regions (**b**) and Hypergeometric Optimization of Motif EnRichment (HOMER) analysis for SREBP2 motif (**c**) in IECs (*n* = 1 mouse per group). **d** Immunoblotting of mouse small intestine and tumor samples. Duo: Duodenum, Jej: Jejunum, Ile: Ileum, Tum: Tumor (*n* = 1 mouse per group). **e** Immunohistochemistry (IHC) for SREBP2 in mouse small intestine and tumors (*n* = 3 mice per group). **f** Immunoblotting of mouse IECs (*n* = 2 tissues from two distinct WT mice, *n* = 3 tissues from three distinct *Prkcz*^f/f^*;Villin-Cre* mice, and *n* = 2 tissues from two distinct *Prkci*^f/f^*;Villin-Cre* mice). **g, h** Inducible deletion of aPKCs in intestinal organoids established from *Prkci*^f/f^*Prkcz*^f/f^*;Villin-Cre*ERT2 mouse small intestine. Schematic and qPCR (**g**) and immunoblotting (**h**) in organoids treated with 1 μM of 4-hydroxytamoxifen (4OHT) or vehicle for 4 days (*n* = 3 biological replicates). **i** Immunoblotting of sgC, sg*PRKCI* and sg*PRKCI/PRKCZ* 293 T cells. **j, k** Immunofluorescence staining of SREBP2 and DAPI (**j**) and nuclear SREBP2/DAPI ratios (**k**) in sgC (*n* = 71 cells

examined over 3 independent experiments and sg*PRKCI* 293 T cells (*n* = 55 cells examined over 3 independent experiments). **l, m** Immunofluorescence staining of SREBP2 and DAPI in WT and *Prkci*^f/f^*Prkcz*^f/f^*;Villin-Cre* mouse small intestine. Representative images (**l**) and SREBP2/DAPI intensity ratio (**m**) in each cell nucleus of WT and *Prkci*^f/f^*Prkcz*^f/f^*;Villin-Cre* mouse small intestinal tumors (WT: *n* = 214 cells from one mouse, Tumor: *n* = 442 cells from one tumor). **n** Dot plot of indicated gene signatures against tumor cell subtypes in the tumor compartment of *Prkci*^f/f^*Prkcz*^f/f^*;Villin-Cre* mouse small intestine (*n* = 5 tumors from two distinct *Prkci*^f/f^*Prkcz*^f/f^*;Villin-Cre* mice). **o, p** Immunofluorescence staining of SREBP2, ANXA10 and DAPI in *Prkci*^f/f^*Prkcz*^f/f^*;Villin-Cre* mouse small intestinal tumors. Representative images (**o**), and ANXA10 intensity in cells with high and low SREBP2 expression (**p**) (cutoff: top and bottom quartile, *n* = 127 cells examined over three different tumor areas from one mouse). Data were presented as mean ± SEM. The ratiometric images of SREBP2 and DAPI signal intensities are shown in the intensity-modulated display mode (IMD) according to the color scale in (**j**) and (**l**). Wald test in **a** HOMER hypergeometric test in **c** and two-tailed, unpaired Student's t-test in **g, k, m** and **p**. Scale bars, 10 μm. Source data are provided as a Source Data file.

(Fig. 5a)[42,43]. It is well established in other systems that, upon the removal of lipids from the culture media, the reduction of intracellular cholesterol levels enhances the translocation of the SCAP-SREBP2 complex resulting in the accumulation of SCAP in the Golgi[42,43]. Notably, our data revealed that PKCλ/ι deficiency resulted in the accumulation of higher levels of SCAP in the Golgi in cells incubated in lipid-depleted media (Fig. 5b, c). Immunoprecipitation experiments demonstrated the interaction between PKCλ/ι and SCAP (Fig. 5d). In addition, SCAP was identified as a substrate of aPKC in an unbiased analog-sensitive screening using the Shokat approach[44] (Supplementary Fig. 5a). Furthermore, in vitro kinase assay with recombinant PKCλ/ι demonstrated that SCAP was a direct phosphorylation substrate of PKCλ/ι (Fig. 5e). Since there is growing evidence that PKCλ/ι regulates the stability of various proteins by direct phosphorylation[16,45,46], we next tested whether the same principle applies to SCAP. Indeed, PKCλ/ι-deficient cells displayed higher SCAP levels than WT cells (Fig. 5f). We next mapped the PKCλ/ι phosphorylation sites in SCAP by crossing the sites identified in the Shokat screening and in silico short sequence motif-based proteome-wide predictions (Scansite 4.0)[47] These analyzes predicted SCAP's T635, S677, and T891 as potential PKCλ/ι phosphorylation sites (Supplementary Fig. 5b). Notably, all these sites were well conserved among different species (Supplementary Fig. 5c), suggesting a crucial role of SCAP phosphorylation by PKCλ/ι in the regulation of SREBP2. To test that hypothesis, we next generated a SCAP mutant in which the three putative PKCλ/ι sites were mutated to alanine (SCAP^AAA). This mutant displayed significantly reduced phosphorylation by PKCλ/ι (Fig. 5g, h). In line with the importance of these sites for SCAP function, cycloheximide (CHX) chase experiments demonstrated the increased stability and half-life of the SCAP^AAA mutant as compared to WT (SCAP^WT) (Fig. 5i, j).

## PKCλ/ι-mediated phosphorylation promotes SCAP degradation by ubiquitination through TRC8

To address the mechanism by which PKCλ/ι regulates the stability of SCAP, we hypothesized that this could be mediated by ubiquitination. Consistent with this assumption, we found that whereas SCAP^WT was ubiquitinated in co-transfection experiments, the ubiquitination of SCAP^AAA was significantly reduced (Fig. 6a). To identify the putative E3-ubiquitin ligase responsible for that effect, we carried out an unbiased screening by proximity-dependent biotinylation assay (BioID2) using PKCλ/ι as the bait[46]. This screening identified interactors of PKCλ/ι corresponding to the REACTOME category "intracellular trafficking from ER to Golgi." Among them, PKCλ/ι was found to bind the ER/COPII proteins SAR1, SEC23, and SEC24, which cluster the SCAP/SREBP2 complex into transport vesicles in the ER[48]; with ERGIC1 and

ERGIC3 from the ERGIC (ER-Golgi intermediate compartment) and with ARF6, ARF1/3, ARF4 (Fig. 6b), which are small GTPases that regulate membrane trafficking at the Golgi[49]. These interactions are consistent with the role of PKCλ/ι in the ER-Golgi trafficking of the SCAP/SREBP2 complex (Fig. 5b). Importantly, along with these organelle transport proteins, the screening identified the ubiquitin ligase TRC8 as a PKCλ/ι interacting partner (Fig. 6b). TRC8 has been previously described as an E3 ubiquitin ligase that might regulate cholesterol metabolism[50,51]. Based on these observations, we posited that SCAP phosphorylation by PKCλ/ι promotes its interaction with TCR8 and its subsequent ubiquitination. Indeed, SCAP ubiquitination was impaired by TRC8 knockdown (Fig. 6c), which significantly increased SCAP stability (Fig. 6d, e).

Next, we evaluated the impact of PKCλ/ι mediated phosphorylation of SCAP on the activity of SREBP2. Akin to the observations in PKCλ/ι-deficient cells, the expression of SCAP^AAA resulted in increased processing and nuclear accumulation of SREBP2 as compared to SCAP^WT (Fig. 6f, g). Immunofluorescence experiments also demonstrated enhanced nuclear SREBP2 in cells stably expressing SCAP^AAA as compared to SCAP^WT (Fig. 6h, i). Nuclear SREBP2 expression was also augmented when TRC8 was knocked down (Fig. 6j, k). These results demonstrate that the PKCλ/ι-mediated phosphorylation of SCAP regulates its stability via the ubiquitin ligase TRC8, which explains the augmented translocation and activation of SREBP2 in aPKC-deficient cells.

## Loss of PKCλ/ι renders cells addicted to cholesterol biosynthesis creating a targetable metabolic vulnerability

One consequence of enhanced cholesterol biosynthesis is that PKCλ/ι deficiency might result in cells becoming less sensitive to extracellular lipid deprivation but addicted to the cholesterol synthetic pathway. Indeed, incubation in LPDS medium resulted in enhanced growth of PKCλ/ι-deficient cells (Fig. 7a, b), which were more sensitive to the incubation with atorvastatin (Fig. 7c–e), a potent inhibitor of the 3-hydroxy-3-methyl glutaryl (HMG)-CoA reductase (HMGCR), a rate-limiting enzyme in the cholesterol biosynthesis pathway[52]. However, cholesterol biosynthesis inhibition with statins can cause a restorative feedback response[53]. Thus, decreased intracellular cholesterol levels promoted by statins induce SREBP2 cleavage and nuclear translocation causing upregulated expression of genes in the cholesterol biosynthesis cascade, negatively impacting the inhibitor effect (Fig. 7f). To shut down this feedback loop and to maximize the therapeutic potency of cholesterol biosynthesis inhibition, we next evaluated the effect of combination therapy of Atorvastatin with Dipyridamole in aPKC-deficient cells. Dipyridamole was recently rediscovered in a drug screening as a potentiator of statins[54], and was shown to block the

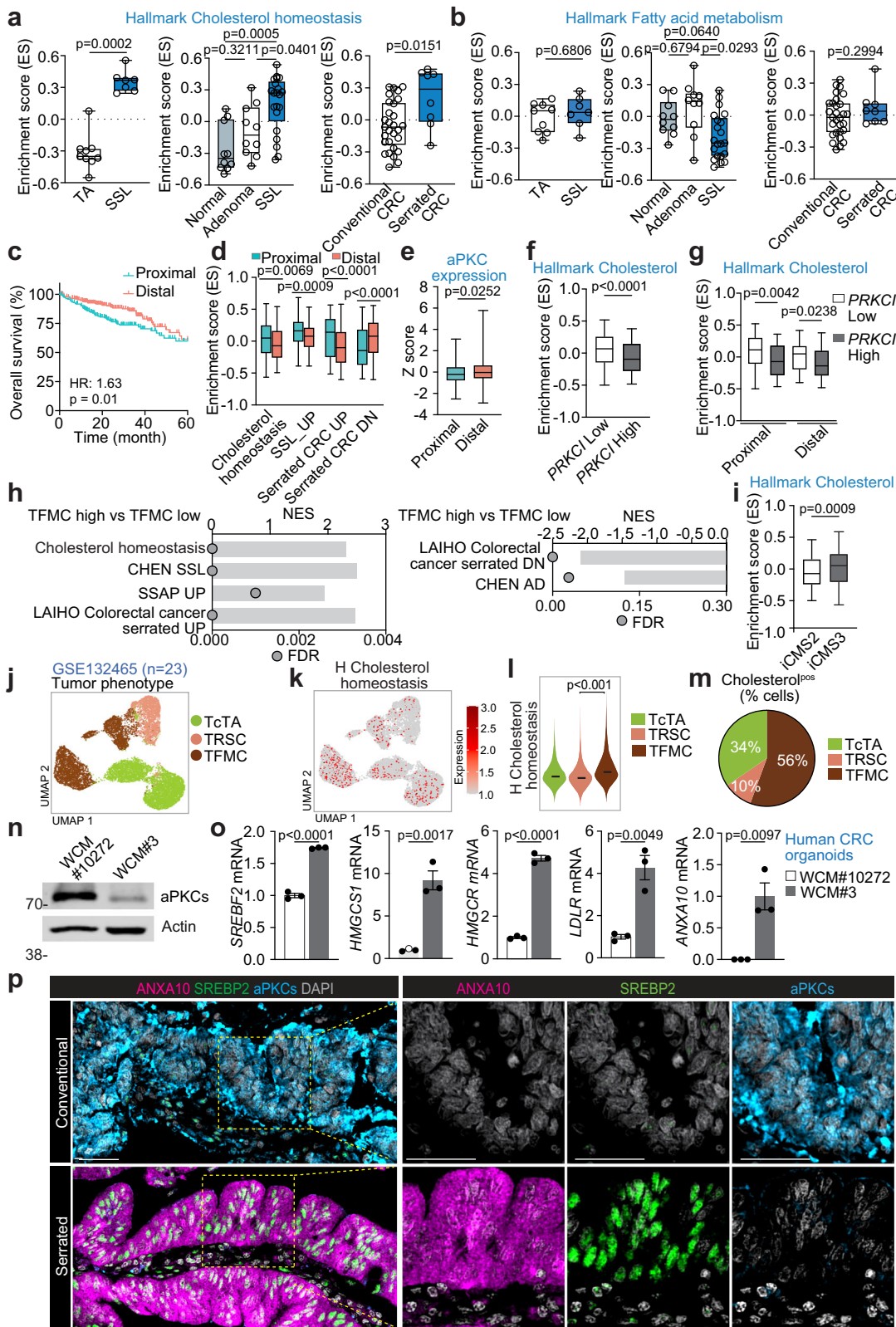

translocation of the SCAP-SREBP complex to the Golgi, thereby blunting SREBP2 activation[55]. Indeed, dipyridamole suppressed the expression of SREBP2 target genes induced by atorvastatin treatment (Fig. 7g, h). Consistently, the addition of Dipyridamole potentiated the Atorvastatin-induced growth inhibition of PKCλ/ι-deficient CRC cells (Fig. 7i). Of translational relevance, human CRC organoids with low

aPKC expression (Fig. 4n) also exhibited addiction to cholesterol inhibition (Fig. 7j). Thus, inhibitors of cholesterol biosynthesis significantly suppressed the growth of aPKC-low human CRC organoids, whereas there was no significant effect on the aPKC-high ones (Figs. 4n and 7j). These results highlight the potential of these inhibitors for treating aPKC-low CRCs.

**Fig. 4 | Dysregulated cholesterol metabolism is associated with low aPKC expression in human serrated tumors. a, b** Enrichment scores (ES) of GSVA on hallmark cholesterol homeostasis (**a**) and fatty acid metabolism in human CRC samples (**b**) in GSE79462 (SSL: $n = 7$, TA: $n = 9$), GSE76987 (Normal: $n = 10$, Adenoma: $n = 10$, SSL: $n = 21$) and GSE4045 (Serrated: $n = 8$, Conventional: $n = 29$). **c** Kaplan–Meier curves for 5-year overall survival of proximal and distal CRC patients in the TCGA COADREAD dataset (Proximal: $n = 205$, Distal: $n = 305$). **d** GSVA enrichment scores of indicated gene sets in proximal ($n = 213$) and distal ($n = 318$) CRC patients in the TCGA. **e** aPKC expression in proximal and distal CRC patients in the TCGA (Proximal: $n = 213$, Distal: $n = 318$). **f** GSVA enrichment scores on hallmark cholesterol in CRC patients with high and low *PRKCI* expression ($n = 148$). **g** GSVA enrichment scores on Hallmark cholesterol in CRC patients stratified by *PRKCI* expression and primary tumor location ($n = 61, 49, 66, 85$). **h** GSEA for TFMC high and low CRC patients in the TCGA ($n = 148$). **i** GSVA enrichment scores on hallmark cholesterol in iCMS2 and iCMS3 CRC patients in the TCGA (iCMS2: $n = 298$, iCMS3: $n = 232$). **j–m** UMAP plot colored by

*Prkci^{f/f}Prkcz^{f/f};Villin-Cre* tumor cell subtypes (**j**), UMAP feature plot colored by the expression of Hallmark cholesterol homeostasis (**k**), violin plot (**l**), and pie chart of the relative distribution of tumor cell subtypes (**m**) expressing the Hallmark cholesterol homeostasis in tumor epithelial cells (GSE132465, $n = 23$ patient samples). **n, o** Immunoblotting (**n**) and qPCR (**o**) in WCM#10272 and WCM#3 human colorectal cancer organoids ($n = 3$ biological replicates). **p** Representative images of immunofluorescence staining for ANXA10, SREBP2, aPKCs, and DAPI in conventional and serrated human CRCs in the tissue microarray (TMA, $n = 468$ patient samples). Data were presented as mean ± SEM. Box and whiskers graphs indicate the median and the 25th and 75th percentiles, with minimum and maximum values at the extremes of the whiskers. Brown-Forsythe and Welch ANOVA tests and Dunnett post hoc test (**a, b**), two-tailed Mann–Whitney test (**a, b, d–g, i**), log-rank test (**c**), one-way ANOVA and post hoc Tukey's test (**g**), two-tailed, unpaired Student's t-test (**l, o**). Scale bars, 50 μm. Source data are provided as a Source Data file.

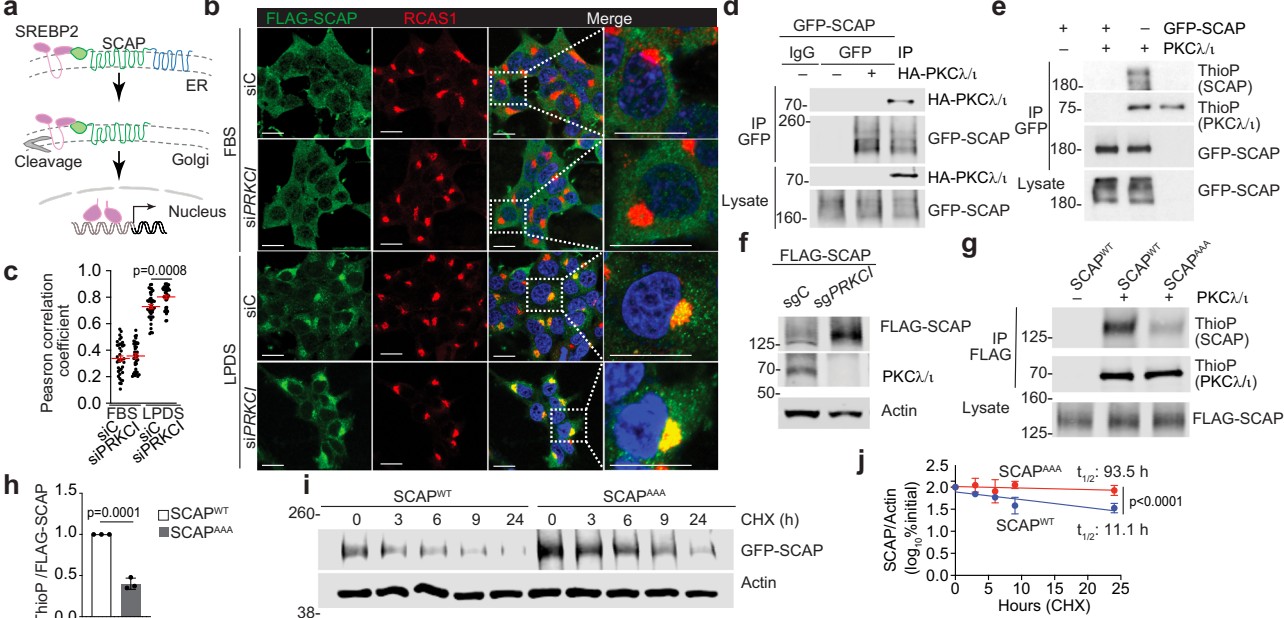

**Fig. 5 | SCAP stability is regulated by PKCλ/ι-mediated phosphorylation.**
**a** Schematic of SREBP2 translocation, cleavage, and activation.
**b, c** Immunofluorescence staining of FLAG, RCAS1, and DAPI in 293 T cells transfected with scramble small interfering RNA (siRNA) (siC) or si*PRKCI* cultured in media containing 10% FBS or 5% LPDS for 24 h. Representative images (**b**) and the Pearson correlation coefficients (**c**) between FLAG and RCAS1 signal intensities (siC-FBS: $n = 37$, si*PRKCI*-FBS: $n = 38$, siC-LPDS: $n = 39$, si*PRKCI*-LPDS: $n = 39$ cells examined over 3 independent experiments). **d** Immunoblotting of cell lysates and GFP-tagged immunoprecipitants of 293 T cells transfected with indicated plasmids. **e** In vitro phosphorylation assay of GFP-tagged SCAP by recombinant PKCλ/ι with ATPγS

followed by p-nitrobenzyl mesylate (PNBM) alkylation and immunoblotting for the indicated proteins. **f** Immunoblotting of cell lysates of sg*PRKCI* and sgC 293 T cells transfected with FLAG-SCAP. **g, h** In vitro phosphorylation assay of FLAG-tagged SCAP WT (SCAP^{WT}) or the triple mutant (SCAP^{AAA}) by recombinant PKCλ/ι. Immunoblotting (**g**) and ThioP/FLAG-SCAP intensity ratio (**h**) ($n = 3$ biological replicates). **i, j** Pulse-chase assay with 50 μg/mL of cycloheximide (CHX) in 293 T cells transfected with SCAP^{WT} or SCAP^{AAA}. Immunoblotting (**i**) and quantification of SCAP intensities normalized to actin (**j**, $n = 3$ biological replicates). Data were presented as mean ± SEM. Two-tailed, unpaired Student's t-test (**c, h, j**). Scale bars, 10 μm. Source data are provided as a Source Data file.

## Cholesterol biosynthesis inhibitors suppress aPKCs-deficient tumors in vivo

To test the impact of cholesterol metabolic reprogramming by aPKC deficiency on tumor development in vivo, *Prkci^{f/f}Prkcz^{f/f};Villin-Cre* mice were fed with regular chow or a diet supplemented with 1.25% cholesterol immediately after the weaning when there was no tumor developed yet (Fig. 8a). The cholesterol-supplemented diet produced a systemic effect significantly elevating serum cholesterol levels in the treated *Prkci^{f/f}Prkcz^{f/f};Villin-Cre* mice (Fig. 8b). More importantly, the cholesterol diet increased the tumor number and load (Fig. 8c, d and Supplementary Fig. 6a), indicating that excessive cholesterol has a tumor-promoting effect in aPKC-deficient CRC.

Given the growth suppressive effect of the combination of atorvastatin and dipyridamole on PKCλ/ι-deficient cells (Fig. 7), we

next investigated the therapeutic potential of this combination therapy on *Prkci^{f/f}Prkcz^{f/f};Villin-Cre* tumors in vivo (Fig. 8e). The complete absence of cleaved SREBP2 in the intestinal tissue (Fig. 8f) and the significant decrease in serum cholesterol levels (Fig. 8g) in these mice confirmed the efficacy of this treatment regimen in blocking cholesterol biosynthesis. Of potential therapeutic relevance, the combination treatment significantly reduced tumor number and load (Fig. 8h–j). This treatment also inhibited malignancy, as determined by a reduction in the percentage of cancer incidence (Fig. 8k) and a reduced number of SSLs and carcinomas (Fig. 8l). Histologically, tumors in the treated mice showed less proliferation, less metaplastic features, and more apoptosis as determined by IHC of Ki67, ANXA10, and cleaved Caspase-3, respectively (Fig. 8m, n).

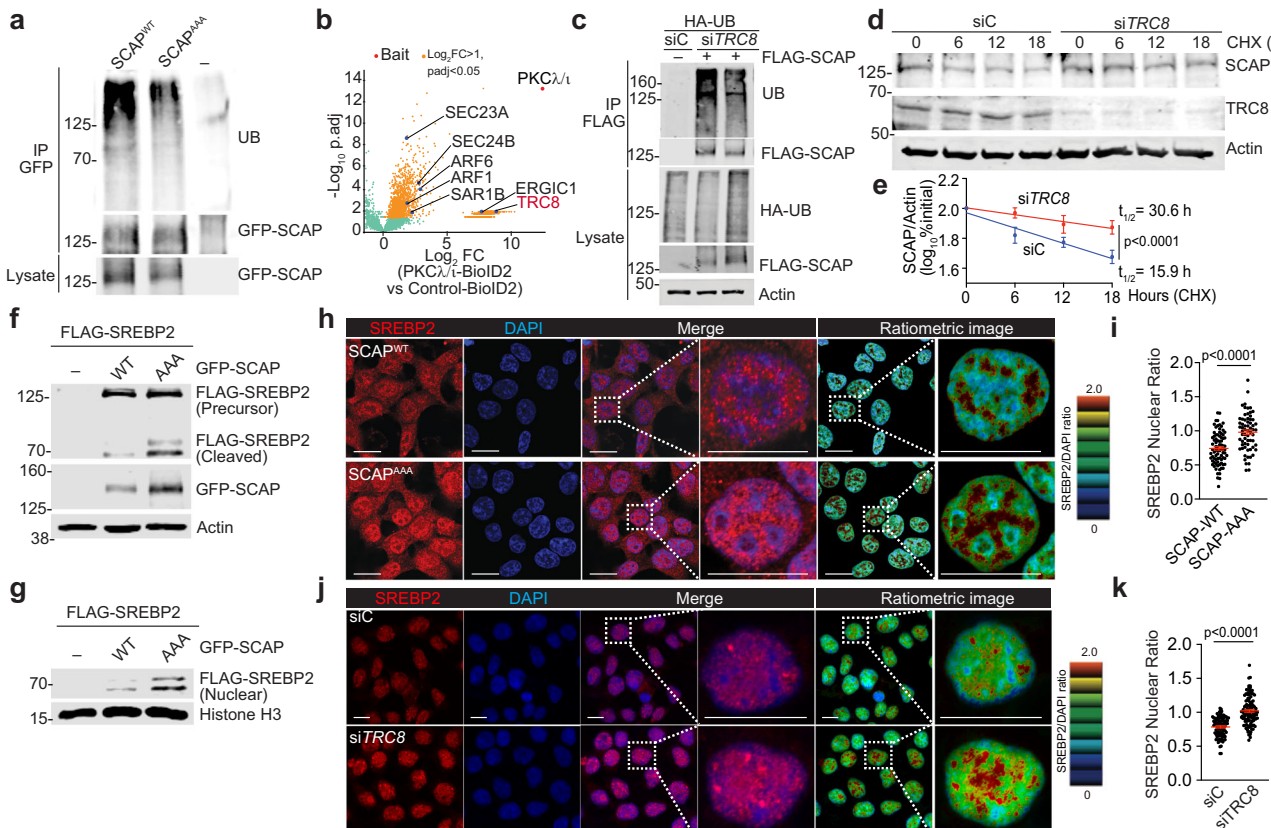

**Fig. 6 | PKCλ/ι-mediated phosphorylation promotes SCAP degradation by ubiquitination through TRC8. a** Immunoblotting of cell lysate and GFP-tagged immunoprecipitates of 293 T cells transfected with indicated plasmids. **b** Volcano plot of biotinylated proteins in PKCλ/ι-BioID2 versus Empty-BioID2 in 293 T cells (*n* = 3 biological replicates). **c** Immunoblotting of cell lysate and FLAG-tagged immunoprecipitates of 293 T cells transfected with indicated plasmids. **d, e** Pulse-chase assay with 50 μg/mL of CHX in 293 T cells transfected with siC or si*TRC8*. Immunoblotting (**d**) and quantification of SCAP intensities normalized to actin (**e**) (*n* = 3 biological replicates). **f** Immunoblotting of 293 T cells transfected and cultured in media containing 5% LPDS for 16 h. **g** Immunoblotting of nuclear fraction of 293 T cells and cultured in media containing 5% LPDS for 16 h.

**h, i** Immunofluorescence staining of SREBP2 and DAPI in 293 T cells stably expressing SCAP^WT and SCAP^AAA cultured in media containing 5% LPDS for 24 h. Representative images (**h**) and nuclear SREBP2/DAPI ratios (**i**) (SCAP^WT: *n* = 82, SCAP^AAA: *n* = 65 cells examined over 3 independent experiments). **j, k** Immunofluorescence staining of SREBP2 and DAPI in 293 T cells transfected with siC or si*TRC8* cultured in media containing 5% LPDS for 24 h. Representative images (**j**) and nuclear SREBP2/DAPI ratios (**k**) (siC: *n* = 98, si*TRC8*: *n* = 110 cells examined over 3 independent experiments). Data were presented as mean ± SEM. Two-tailed, unpaired Student's t-test. Scale bars, 10 μm. Source data are provided as a Source Data file.

These observations were further validated by multiplex staining of the tumors (Fig. 8o). Thus, the majority of metaplastic (ANXA10-positive) tumor cells also showed high nuclear SREBP2 signals in the vehicle-treated mice, and the cholesterol inhibition treatment significantly decreased the expression of both ANXA10 and SREBP2. Considering that TFMCs predict poor survival, exhibited the highest expression of cholesterol signatures (Fig. 1p), and that cholesterol metabolic reprogramming rendered cells addicted to cholesterol inhibition (Fig. 7), these results suggest that this treatment reduced tumor development and its malignant potential by effectively suppressing the growth of TFMCs in *Prkcι^f/f Prkcz^f/f;Villin-Cre* tumors. Consistently, similar results were obtained with the small molecule fatostatin, which inhibits the trafficking of SCAP-SREBP2 complex from the ER to the Golgi (Supplementary Fig. 6b)[56]. Fatostatin treatment reduced intestinal SREBP2 cleavage and serum cholesterol levels (Supplementary Fig. 6c, d), as well as tumor number and load (Supplementary Fig. 6e, f). In summary, the upregulation of cholesterol biosynthesis in aPKC-deficient intestinal epithelial cells is an important contributor to serrated malignancy but also renders cancer cells addicted to cholesterol synthesis, creating an actionable therapeutic vulnerability for this type of very aggressive CRC (Fig. 8p).

## Discussion

Our study here unveils a previously unappreciated upregulation of the cholesterol biosynthetic pathway driven by low aPKC expression in CRC. This new evidence has to be considered in the context of our recent research on the role of the aPKCs, and particularly PKCλ/ι, as tumor suppressors by directly phosphorylating key substrates in cancer and regulating their stability[16,45]. Thus, in the intestinal tumorigenesis paradigm, we previously reported that PKCλ/ι phosphorylates EZH2 and ULK2 thereby regulating apoptosis, autophagy, and inflammation[46,57]. Our present study demonstrates that SCAP is a direct substrate for PKCλ/ι, and that the loss of PKCλ/ι and the subsequently increased SCAP stability results in the enhanced activation of SREBP2, the master regulator of cholesterol biosynthesis[58]. SCAP phosphorylation recruits the ER-bound TCR8 that promotes SCAP ubiquitination and degradation. Few prior reports have addressed the regulation of SCAP levels and activity by post-translational modifications, including phosphorylation. SCAP has been shown to be phosphorylated by the cAMP/PKA pathway, which promoted SREBP2 processing[59], in contrast to PKCλ/ι-phosphorylation that represses its activity. Two ER membrane ubiquitin ligases, RNF145 and RNF5, have been reported to induce SCAP ubiquitination with different outputs in

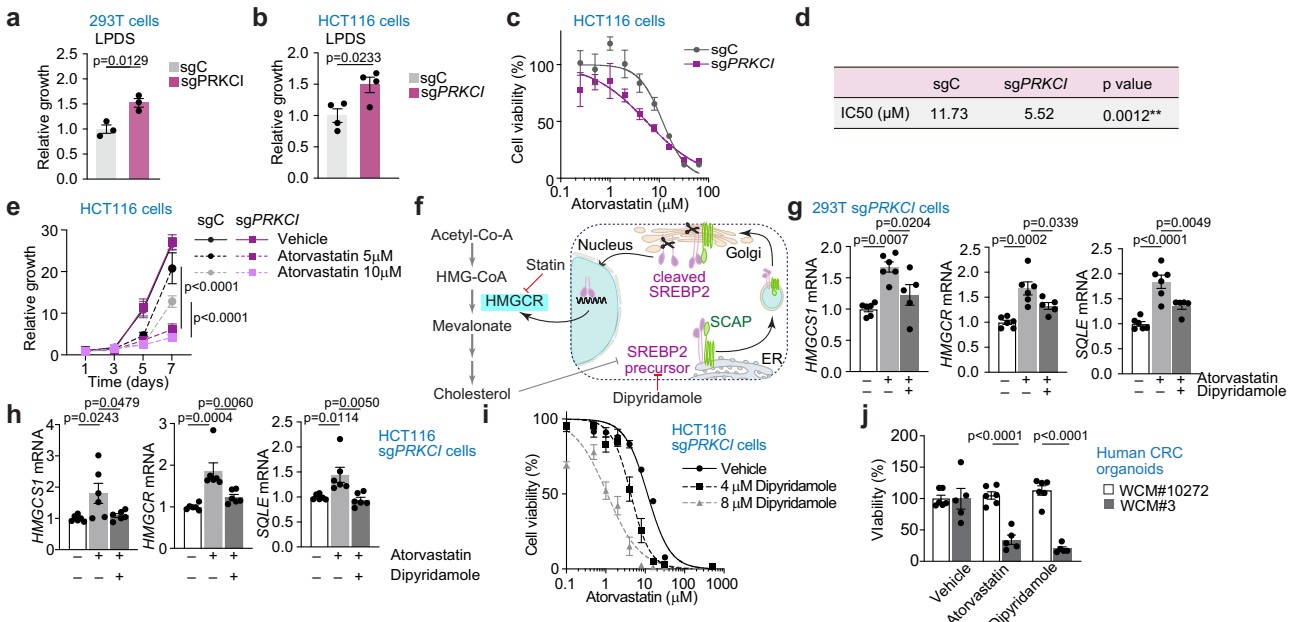

**Fig. 7 | Loss PKCλ/ι of renders cells addicted to cholesterol biosynthesis.**
**a** Relative growth of sgC and sg*PRKCI* 293 T cells cultured in media with 5% LPDS for
4 days (*n* = 3 biological replicates). **b** Relative growth of sgC and sg*PRKCI* HCT116
cells cultured in media with 5% LPDS for 4 days (*n* = 4 biological replicates).
**c**, **d** Dose response curves (**c**) and IC50 (**d**) of atorvastatin in sg*PRKCI* and sgC
HCT116 cells (*n* = 6 biological replicates per group). **e** Growth curves of sgC and
sg*PRKCI* HCT116 cells treated with vehicle or indicated concentration of atorvas-
tatin (*n* = 4 biological replicates). **f** Schematic of cholesterol biosynthesis pathway
and feedback response caused by statin treatment. **g** qPCR of sg*PRKCI* 293 T cells
treated with 10 μM of atorvastatin and/or 5 μM of dipyridamole for 16 h (-/-, +/-:

*n* = 6 biological replicates, +/+: *n* = 5 biological replicates). **h** qPCR of sg*PRKCI*
HCT116 cells treated with 10 μM of atorvastatin and/or 5 μM of dipyridamole for
16 h (*n* = 6 biological replicates). **i** Dose-response curves of atorvastatin with vehicle
or indicated concentration of dipyridamole in sg*PRKCI* HCT116 cells (*n* = 5 biolo-
gical replicates per group). **j** Relative cell growth of human CRC organoids treated
with vehicle, 10 μM of atorvastatin, or 10 μM of dipyridamole for 4 days
(WCM#10272: *n* = 6 biological replicates, WCM#3: *n* = 5 biological replicates). Data
were presented as mean ± SEM. Two-tailed, unpaired Student's t-test (**a**, **b**), Two-
tailed, extra sum-of squares F test (**d**), one-way ANOVA and post hoc Tukey's test
(**e**, **g**, **h**, **j**). Source data are provided as a Source Data file.

cholesterol biosynthesis but without affecting SCAP abundance or
stability[60,61].

The interrogation of a cohort of human CRC specimens supports
the relevance of our findings. Thus, we found that SREBP2 activation is
associated with serrated tumorigenesis and reduced levels of the
aPKCs, which also correlates with metaplastic markers such as ANXA10
in human samples. Therefore, the biochemical link reported here
between PKCλ/ι and the activation of SREBP2 through the regulation of
SCAP stability and the relevance of this pathway for TFMC explains, at
a mechanistic level, the role of cholesterol in metaplastic activation
and aggressive serrated tumor initiation and evolution. These results
suggest that the tumor cell of origin of serrated tumors might inher-
ently harbor high-ANXA10/low-aPKC expression features and an ele-
vation of the mevalonate pathway. However, additional molecular
mechanisms beyond the direct phosphorylation of SCAP may also
contribute to the dysregulation of the cholesterol pathway in these
tumors. Thus, the loss of PKCλ/ι upregulates MYC targets in the
intestinal epithelium (Supplementary Fig. 2), which have been shown
to transcriptionally activate cholesterol biosynthesis[62].

The upregulated cholesterol biosynthesis in this type of aPKC-
deficient tumor cells creates a metabolic addiction, which is also a
therapeutic vulnerability. Thus, statin-based inhibitors strongly sup-
press tumor growth in various experimental settings, including
endogenous mouse tumor models, genetically engineered cultured
cancer cells, and patient-derived CRC organoids. Detailed analysis of
serrated tumors in the aPKC-deficient mice further revealed in our
study that among the different tumor cell subtypes, the TFMCs exhibit
the highest levels of expression of the cholesterol signatures and that
cholesterol inhibition treatment preferentially ablates these TFMCs.
This is of particular importance because TFMCs have been shown to
predict poor survival in CRC and have been proposed to be key players

in orchestrating the desmoplastic and immune-excluded micro-
environment that accounts for the aggressive phenotype of
mesenchymal serrated tumors[14,15]. Therefore, an important corollary of
these observations is that targeting cholesterol biosynthesis can be an
effective way to reduce the aggressiveness of mCRC by removing
the TFMCs.

Accurate disease stratification and optimal patient selection are
essential for future clinical application. The role of cholesterol
metabolism remains inconclusive at the epidemiological level in the
overall CRCs, in part because of the heterogeneous nature of the
disease. However, given the association between serrated tumor-
igenesis and the cholesterol biosynthetic pathways, efforts toward
better stratification of CRC patients have been hindered by the dif-
ficulty to identify serrated CRCs, which can lose their histological
features, especially at advanced stages[4]. Thus, while early serrated
lesions are histologically different and easily differentiated from
conventional pre-neoplasias[6], however, in the advanced carcinoma
stage, the serrated histological criteria are less easily identifiable
and, therefore, in need of better biomarkers, ideally linked to sig-
naling and metabolic drivers that could be therapeutically targeted.
According to our data, high ANXA10 expression, a marker for the
TFMCs, strongly correlates with nuclear SREBP2 signals in human
patients. These results indicate that SREBP2, a potentially actionable
target, could be used along with low aPKC and high ANXA10/MUC5A
as biomarkers for the identification and selective treatment of ser-
rated CRC subtypes that, although they might be refractory to
conventional therapy, would be addicted to the cholesterol path-
way, creating a potential clinical vulnerability. Future case-control
studies or prospective cohort studies would be warranted to
investigate the association between cholesterol metabolic profiles
and the epidemiology of serrated CRCs stratified by these markers.

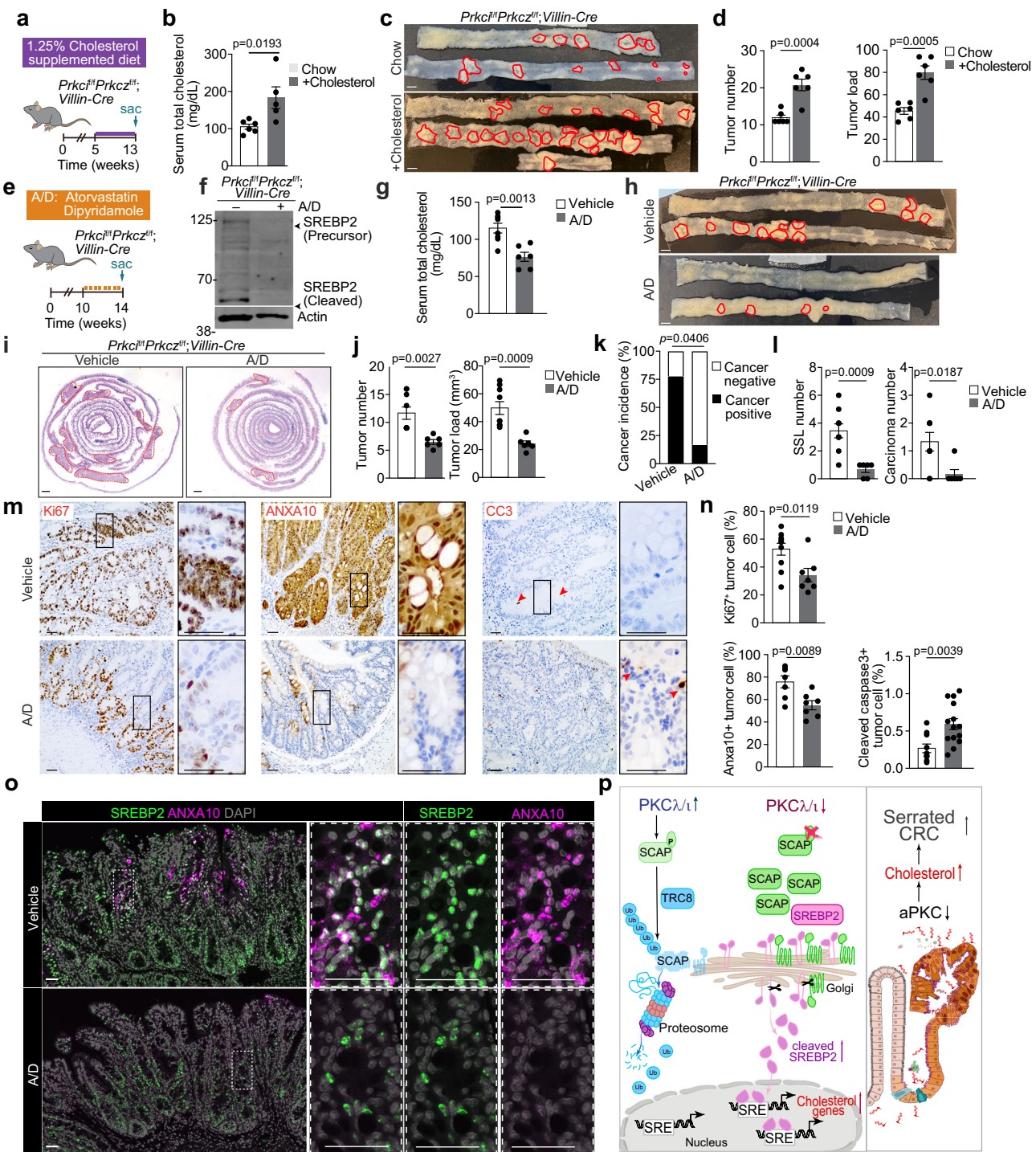

**Fig. 8 | Tumor suppressive effects of cholesterol inhibition in aPKC-deficient serrated tumors. a–d** *Prkci[f/f]Prkcz[f/f];Villin-Cre* mice fed with regular chow or 1.25% cholesterol-supplemented diet for 3 months. Experimental design (**a**), serum total cholesterol levels (**b** Chow:*n* = 6 mice, Cholesterol:*n* = 5 mice), macroscopic images (**c**), total tumor number, and tumor load (**d**) Chow:*n* = 6 mice, Cholesterol:*n* = 6 mice). Red lines denote macroscopic tumors in (**c**). **e–l** *Prkci[f/f]Prkcz[f/f];Villin-Cre* mice treated with vehicle (*n* = 9) or 50 mg/kg of atorvastatin orally and 120 mg/kg of dipyridamole intraperitoneally (A/D) (*n* = 6) daily for 4 weeks. Experimental design (**e**), immunoblotting of small intestines (**f**), serum total cholesterol levels (**g**) macroscopic images (**h**), H&E staining of tumors (**i**), total tumor number and tumor load (**j**), cancer incidence (**k**), and numbers of sessile serrated lesions (SSL) and carcinomas (**l**). Red lines denote macroscopic tumors in **h** and microscopic tumor areas in (**i**). **m, n** Representative images of Ki67, ANXA10, and Cleaved Caspase 3 (CC3), staining (**m**) and their quantification (**n**) (Ki67, vehicle: *n* = 10 fields examined from

3 mice, A/D: *n* = 7 fields examined from 2 mice; ANXA10, vehicle: *n* = 7 fields examined from 2 mice, A/D: *n* = 7 fields examined from 2 mice; CC3, vehicle: *n* = 10 fields examined from 2 mice, A/D: *n* = 14 fields examined from 2 mice). Red arrows denote positive cells for CC3. **o** Representative images of immunofluorescence staining for SREBP2, ANXA10, and DAPI in vehicle and A/D treated tumors. **p** Graphical scheme of PKCλ/ι mediated cholesterol metabolic reprogramming in the serrated tumorigenesis. In the physiological state, PKCλ/ι directly phosphorylates SCAP promoting degradation through a ubiquitination cascade regulated by TRC8. Upon the loss of PKCλ/ι the subsequently increased SCAP stability results in the enhanced activation of SREBP2, the master regulator of cholesterol biosynthesis, to promote serrated CRC development. Data were presented as mean ± SEM. Two-tailed, unpaired Student's t-test (**b, d, g, j, l, n**), two-tailed chi-square test (**k**). Scale bars, 5 mm (**c, h**), 1 mm (**i**), 50 μm (**m, o**). Source data are provided as a Source Data file.

Another approach toward clinical application is maximizing therapeutic efficacy. Statins have been widely used for decades to treat hypercholesterolemia and are generally safe and well tolerated in patients with impaired liver or renal functions, which are the common comorbidities in cancer[63]. However, one of the critical drawbacks of statin treatment as anticancer therapy is the SREBP2-mediated feedback response[64]. Reduced intracellular cholesterol levels caused by statin administration induce SREBP2 activation and transcription of cholesterol pathway genes blunting treatment efficacy. Furthermore, SREBP2 also upregulates the expression of other genes, such as LDLR, which restores intracellular cholesterol levels by increasing its uptake from extracellular sources[64]. One way to overcome this feedback loop is to find ways that can potentiate the effects of statins, especially by suppressing SREBP2 activation. In this regard, we have employed Dipyridamole in combination with Atorvastatin, which demonstrates strong anticancer activity in aPKC-deficient cells and tumors. While efforts have been made to identify the combinations that synergize the statin effect in various malignancies, including leukemia, melanoma, and breast cancer[53,54,65,66], potential new efforts using colorectal cancer cells with low aPKC expression are needed to discover new drugs exploiting other signaling cascades activated by aPKC deficiency to synergize with cholesterol-targeted therapies. For example, our previous study[15] demonstrated that targeting the enhanced hyaluronan levels in serrated tumors reduces the TFMC population and reprograms the stroma and the immune microenvironment making these tumors vulnerable to immunotherapy. Since TFMCS are addicted to cholesterol biosynthesis, one can envision combinatorial therapy for serrated tumors using cholesterol-targeting drugs together with immune checkpoint blockade. Also, recent data support the notion that increased cholesterol metabolism is critical in BRAF-driven serrated carcinogenesis, which suggests that patients harboring BRAF mutations might benefit from cholesterol-lowering therapies[67].

In addition to the critical role of cholesterol metabolism in serrated cancer, our results have revealed that cholesterol metabolism is also dysregulated in human precancerous SSLs and non-tumoral regions of the aPKC-deleted mouse intestine. Given that SSLs possess distinct molecular features compared to conventional tubular adenomas at the single cell level[68], preneoplastic lesions in the serrated pathway may already be addicted to cholesterol biosynthesis, opening a new avenue for the chemoprevention of serrated CRCs. More detailed analyzes of large-scale datasets for SSLs should provide a better understanding of the role of cholesterol metabolism at the premalignant stage. This is important because there is no indication of chemoprevention focusing on specific CRC subsets so far. Aspirin is currently the only agent approved for that end, but it is not a tailored approach to specific CRC subtypes, and its use is limited to young patients at high risk of cardiovascular disease and has several adverse effects, such as gastrointestinal and intracranial bleeding[31]. Therefore, as discussed above, pharmacological evidence from patient-derived SSL organoids and appropriate patient selection with aPKC, ANXA10/MUC5A, and SREBP2 supports future randomized controlled trials for serrated CRC chemoprevention.

Finally, the precise mechanism by which the cholesterol pathway regulates tumorigenesis can be multifaceted[20–22]. Cholesterol biosynthesis is not only responsible for cholesterol as an end product but also creates isoprenoids as intermediate metabolites that can be the source of geranylgeranyl pyrophosphate (GGPP) and ubiquinone, contributing to protein prenylation, glycosylation, and the response to oxidative stress[21,24]. However, the tumor-promoting effects of the cholesterol diet observed in the experiments reported here in aPKC-deficient serrated tumors as well as in other preclinical disease models suggests that cholesterol per se rather than intermediate metabolites plays a central role[25,69]. In addition to serving as a precursor of anabolic steroid hormones, cholesterol also mediates multiple cellular signaling pathways[70]. For instance, cholesterol has been reported to stabilize and upregulate TAZ via the adenyl cyclase-RhoA pathway[71]. YAP/TAZ pathway plays a pivotal role in physiological stem cell homeostasis and tumorigenesis, and which has also been upregulated in aPKC-deficient tumors[14,72,73]. Thus, the aggressive phenotypes observed in aPKC-deficient cancer cells are most likely driven by the pathways that are activated by excessive intracellular cholesterol.

In summary, we have discovered a regulatory mechanism in cholesterol metabolism governed by aPKCs. This aPKC-ANXA10-SREBP2 axis establishes a paradigm for metabolic reprogramming, which provides potential opportunities for therapeutic intervention and prevention in the context of serrated mCRC, a very aggressive type of neoplasia affecting a large proportion of patients with poor prognosis and in need of more tailored and efficient therapies.

## Methods

This study was approved by the IRB Committee of Weill Cornell Medicine, the Osaka City University Ethics Committee, and the HGUSL ethical board, and written informed consent was obtained from the patients.

### Human samples and ethics approval

For CRC samples, surgically resected were obtained from males ($n = 194$, age median:66.1 years) and females ($n = 189$, age median:67.8 years) at Osaka Metropolitan University Hospital, and human CRC samples from males ($n = 50$, age median:70.0 years) and females ($n = 50$, age median:69.9 years) at Santa Lucía General University Hospital (HGUSL), Spain. CRC tissues were obtained from each patient, and none of them had undergone preoperative radiation or chemotherapy. This study was approved by the Osaka City University Ethics Committee and the HGUSL ethical board, and written informed consent was obtained from the patients. De-identified human samples were sent to Weill Cornell Medicine and used for histological analyzes. The study was approved by the IRB Committee of Weill Cornell Medicine. This study does not involve any race, ethnicity, or socially relevant category considered for the analysis. CRC organoids were generated from adult patients with disease diagnoses using surgical resection or endoscopic biopsy as starting material and were de-identified by the research team. Patients were recruited for organoid generation in 2021 at Weill Cornell Presbyterian Hospital with informed consent under the approval of the IRB Committee of Weill Cornell Medicine.

### Mice

*Prkci*$^{f/f}$, *Prkcz*$^{f/f}$, *Prkci*$^{f/f}$*Prkcz*$^{f/f}$, Villin-Cre, *Villin-Cre*ERT2, *Prkci*$^{f/f}$;*Villin-Cre*, *Prkcz*$^{f/f}$;*Villin-Cre*, *Prkci*$^{f/f}$*Prkcz*$^{f/f}$;*Villin-Cre*, and *Prkci*$^{f/f}$*Prkcz*$^{f/f}$;*Villin-Cre*ERT2 mice were previously described[14,15,57,74–77]. All mouse strains were generated in a C57BL/6 background and were born and maintained under pathogen-free conditions. For cholesterol biosynthesis inhibition, 10-week-old *Prkci*$^{f/f}$*Prkcz*$^{f/f}$;*Villin-Cre* mice were treated with a combination of oral atorvastatin (Selleckchem Cat# S5715; 50 mg/kg) and intraperitoneal dipyridamole (Selleckchem Cat# S1895; 120 mg/kg) daily for 4 weeks. Control mice were treated with vehicle. For fatostatin treatment experiments, 10-week-old *Prkci*$^{f/f}$*Prkcz*$^{f/f}$;*Villin-Cre* mice were treated with intraperitoneal fatostatin (Selleckchem Cat# S8284; 15 mg/kg) daily for 4 weeks. To induce CreERT2-mediated recombination, 8- to 12- week-old mice received one dose of 6 mg of tamoxifen (Millipore-Sigma Cat# H7904) in corn oil by intraperitoneal injection. For cholesterol-supplemented diet experiments, 4-6-week old *Prkci*$^{f/f}$;*Villin-Cre*, *Prkci*$^{f/f}$*Prkcz*$^{f/f}$;*Villin-Cre*, or WT control mice were fed with a regular chow diet (PicoLab Rodent Diet, #5053) or 1.25% cholesterol supplemented diet (Research Diets, #C22021701i) for 2 months. Animal handling and experimental procedures conformed to institutional guidelines and were approved by the Sanford-Burnham-Prebys Medical Discovery Institute Institutional Animal Care and Use Committee and by the Weill Cornell Medicine Institutional Animal Care and Use

Committee. The mice were maintained in a 14 h light / 10 h dark cycle, and the housing temperature and humidity were 24 °C and 50%, respectively. All mice were maintained on food and water ad libitum and were age-matched and co-housed for all experiments. Mice were sacrificed and small intestine, tumors, or other organs were collected for analysis. All genotyping was done by PCR. Age- and sex-matched mice were used for all experiments. Cholesterol-related phenotypes, tumor growth, and response to treatment were comparable between the sexes. Overall numbers are as follows. Villin-Cre-Prkcz-/- Prkci-/- (male: $n = 20$, female: $n = 40$), C57BL/6 (male: $n = 10$, female: $n = 10$), NSG (male: $n = 2$, female: $n = 2$). The end-point permitted by the ethics committee was 20% of body weight loss. We ensured that each time mice were sacrificed, the maximal body weight loss did not exceed this limit.

## Cell lines

For intestinal epithelial cell (IEC) isolation, small intestines were removed, washed with cold PBS, opened longitudinally, cut into 5 mm fragments, and washed several times with cold PBS until clean. Cut tissue fragments were incubated with 5 mM EDTA in PBS containing 10% FBS for 40 min at 4 °C. IECs were then mechanically separated from the connective tissue by vigorous shaking and then filtered through a 100 μm mesh into a 50 mL conical tube to remove tissue fragments. Isolated IECs were pelleted and then lysed for RNA or protein extraction. Patient-derived organoids (PDOs) were established from patient-derived xenografts (PDXs). First, surgically resected colorectal cancer tissues were transplanted into 4-6 week-old NSG mice (NOD.Cg-Prkdc$^{scid}$ Il2rg$^{tm1Wjl}$/SzJ, The Jackson Laboratory, strain: 005557) to generate PDXs. After engraftment, PDX tissues were washed vigorously with ice-cold PBS and minced into 1 mm$^3$ fragments using surgical scalpel blades. The fragments were digested with a digestion buffer containing 25 mg/mL Collagenase A, 25 mg/mL Dispase II, and 500 u/mL DNase I for 10 min at 37 °C. The supernatant was filtered through a 70 μm cell strainer and centrifuged at 400 g for 5 min. The pellet was embedded in Matrigel and applied onto 12-well plates as 20 μL droplets. Organoids were cultured in advanced DMEM/F12 supplemented with 50% Wnt-3A conditioned medium, 50 ng/mL recombinant human EGF, 100 ng/mL recombinant murine Noggin, 500 ng/mL recombinant murine R-Spondin 1, 500 nM A83-01, 100 ng/mL recombinant human IGF-1, 50 ng/mL recombinant human FGF-2, 10 μM Y-27632, 1x B27 supplement, 10 nM human [Leu15]-Gastrin I, 1x GlutaMAX, 1x, and 1x penicillin/streptomycin. After the first passage, Wnt-3a and R-spondin 1 were removed from the media. For the induction of Cre-mediated recombination in organoids, 500 nM 4-OH tamoxifen was added to the media for the duration of each experiment. Mouse tumor organoids (MTOs) were obtained from Dr. Eduard Batlle (Institute for Research in Biomedicine, Barcelona, Spain)[37]. MTOs were cultured in advanced DMEM/F12 medium supplemented with 10 mM HEPES, 1x GlutaMAX, 1x B27 supplement, 50 ng/mL recombinant human EGF. Normal intestinal organoids were cultured in Advanced DMEM/F12 containing 10 mM HEPES, 1X GlutaMAX, 1X N2 supplement, 1X B27 supplement, 50 ng/mL EGF, 500 ng/mL R-Spondin 1, 100 ng/mL Noggin, and 10 μM Y-27632. 293 T (CRL-3216, sex: female), HCT116 (CCL-247, sex: male), and L Wnt-3A (CRL-2647, sex: male) cell lines were obtained from ATCC. Cells were cultured in DMEM supplemented with 10% FBS, 1% glutamine and 1% penicillin/streptomycin. Organoids and cell lines were cultured in an atmosphere of 95% air and 5% $CO_2$ at 37 °C and tested monthly for mycoplasma contamination.

## Cell culture experiments

293 T sg*PRKCI*, MTO sg*Prkci*, and MTO *sgPrkci/Prkcz* were described previously[15,46,57]. To knock out *PRKCI* and *PRKCZ* in 293 T cells and HCT116 cells, single-guide RNA sequences targeting *PRKCI* and *PRKCZ* purchased from Synthego (guides are described in Supplementary

Table 2) were transduced with recombinant Streptococcus pyogenes Cas9 protein, using the Neon Transfection System 1 (Invitrogen) following the manufacturer's protocol. Single clones were expanded and screened by protein immunoblotting. Knockdown of *PRKCI* and *TRC8* genes in 293 T cells were achieved by siRNA transfection using Lipofectamine RNAiMAX Transfection Reagent (Invitrogen), siRNA oligonucleotides sequences are described in Supplementary Table 3. Transient overexpression was achieved by transfection using X-tremeGENE HP transfection reagent (Roche). Transfected cells were examined 48 h after transfection. Point mutations to disrupt the predicted phosphorylation sites in SCAP were introduced by Site-Directed Mutagenesis (Stratagene). Stably expressing SCAP$^{WT}$ and SCAP$^{AAA}$ cells were established by lentivirus-mediated transduction. Lentiviruses were produced in 293 T cells using X-tremeGENE HP transfection reagent (Roche). Virus-containing supernatants were collected 48 h after transfection, filtered to eliminate cells, and supplemented with 8 μg/mL polybrene. Cells were infected with viral supernatants and selected with puromycin (3 μg/mL). For cell growth assays, cells were seeded in 384-well plates at 500 cells. For dose-response assays, cells were seeded in 384-well plates at 1000 cells and subjected to various treatments for 4 days. Cell viability was assessed by nuclear acid staining with SYTO60 (Invitrogen). After fixation with 4% PFA for 10 min at room temperature, cells were stained with 1 μM of SYTO60 for 60 min, then the fluorescence was measured by an Infrared imager (LI-COR) using 700 nm channel. Cell viability in organoids was assessed by CellTiter Glo 3D. Dose-response curves and IC50s were determined by nonlinear regression.

## Histology, immunohistochemistry, and immunofluorescence

Tissues were fixed overnight in 10% neutral buffered formalin and embedded in paraffin. Paraffin-embedded tissues were cut (5 μm) and stained with hematoxylin and eosin (H&E). For immunostaining, paraffin sections were subjected to antigen retrieval and blocked in Protein Block Serum-Free solutions (DAKO). Incubation with primary antibodies was performed overnight at 4 °C, followed by incubation with biotinylated secondary antibodies and visualized with avidin-biotin complex using diaminobenzidine as the chromogen. Incubation in 3% $H_2O_2$ for 10 min at room temperature was used to quench endogenous peroxidase. Immunostaining was analyzed with a Zeiss light microscope supplemented with Zen 3.3 Blue edition software. For the multiplex immunofluorescence, the OPAL™ 4-Color Manual IHC Kit (NEL810001KT; Akoya Biosciences) was used with fluorophores Opal 520, Opal 570, and TSA Plus Cyanine 5 (NEL745001KT; Akoya Biosciences), and Spectral DAPI counterstaining. For immunocytochemistry, after fixation, permeabilization, and blocking, cells were incubated with primary antibody overnight at 4 °C, followed by fluorophore-conjugated secondary for 2 h and counterstaining with DAPI. Coverslips were mounted and imaged under confocal microscopy. The following primary antibodies were used for immunostaining: SREBP2 (1:1000 in IHC, 1:200 in the OPAL system, 1:100 in immunocytohistory, ab30682; Abcam), Ki67 (1:200, 12202; Cell Signaling Technology), ANXA10 (1:1000 in IHC and the OPAL system, ab213656; Abcam), Cleaved Caspase-3 (1:100, 9664; Cell Signaling Technology), aPKCs (1:100, ab59364; Abcam), FLAG (1:500, F3165; Sigma-Aldrich). For the OPAL system, the secondary antibody used was either Anti-Ms + Rb HRP (NEL810001KT; Akoya Biosciences) or Rabbit IgG, secondary, HRP (1:300, 31461; Thermo Fisher Scientific), for human and mouse tissues, respectively. For immunocytochemistry, the following secondary antibodies were used 1:200: Mouse IgG1, Alexa Fluor 488 (A21121; Thermo Fisher Scientific), Donkey anti-Rabbit IgG, Alexa Fluor 568, (A10042; Thermo Fisher Scientific). Acquired images were analyzed with QuPath and ImageJ. Ratiometric images of SREBP2 and DAPI signal intensities (SREBP2/DAPI) in the nucleus were visualized in the intensity-modulated display mode (IMD) by using an ImageJ plugin[78], in which eight colors from red to blue represent the

SREBP2/DAPI ratio and the 32 grades of color intensity represent the DAPI intensity according to the color scale.

## Tissue microarray analyses

Scanned TMA slides were dearrayed and preprocessed using QuPath (v.0.4.0). After dearraying, all cores were examined except for those without any tumors or with artifacts. Cell-detection was determined for each core, and the total number of positive cells was assessed using DAPI staining as a reference. CRC patients were categorized into high and low SREBP2, aPKCs, or ANXA10 expression based on H-scores. Serrated CRCs were diagnosed by experienced pathologists with H&E-stained sections according to the criteria previously described[79–81].

## Immunoblotting analysis

Cells for protein analysis were lysed in RIPA buffer (20 mM Tris-HCl, 37 mM NaCl2, 2 mM EDTA, 1% Triton-X, 10% glycerol, 0.1% SDS, and 0.5% sodium deoxycholate) with phosphatase and protease inhibitors. For immunoprecipitations, cells were lysed in IP lysis buffer (100 mM NaCl, 25 mM Tris, 1% Triton-X, 10% glycerol, with phosphatase and protease inhibitors) and immunoprecipitated with the following antibodies at 4 °C overnight: GFP (1:100000, ab290; Abcam), Rabbit IgG (1:10000, 2729; Cell Signaling Technology), Mouse IgG (1:10000, sc-2025; Santa Cruz Biotechnology). Immunoprecipitated proteins were collected with 25 μL of 50% slurry of Sepharose beads (Invitrogen) and were washed three times with lysis buffer, once with high salt (500 mM NaCl), and once more with lysis buffer. Nuclear and cytoplasmic fraction was extracted by NE-PE Nuclear and Cytoplasmic Extraction Reagents (Thermo Scientific) following the manufacturer's protocol. Protein concentration in lysates were determined by using Protein Assay Kit (Bio-Rad). Cell extracts were denatured, subjected to SDS-PAGE, transferred to PVDF membranes (GE Healthcare). After blocking with blocking buffer (LI-COR), the membranes were incubated with the following antibodies overnight at 4 °C: aPKCs (1:1000, ab59364; Abcam), PKCλ/ι (1:500, 610208; BD Biosciences), TRC8 (1:500, sc-390347; Santa Cruz Biotechnology), SREBP2 (1:1000, ab30682; Abcam), GFP (1:1000, 2956; Cell Signaling Technology), Thiophosphate ester (1:5000, ab92570; Abcam), SCAP (1:500, sc-13553; Santa Cruz Biotechnology), SCAP (1:500, PA5-28982; Thermo Fisher Scientific), Ubiquitin (1:2000, sc-8017; Santa Cruz Biotechnology), FLAG (1:2000, F3165; Sigma-Aldrich), HA-Tag (1:1000, 3724; Cell Signaling Technology), β-actin (1:40000, A1978; Sigma-Aldrich). After 2 h incubation with the appropriate fluorochrome-conjugated secondary antibodies 1:3000 (Rabbit IgG, IRDye 800, 926-32211; LI-COR Biosciences, Mouse IgG1, IRDye 800, 926-32350; LI-COR Biosciences, Mouse IgG, IRDye 800, 926-32210; LI-COR Biosciences), the immune complexes were detected by Near-infrared fluorescence (LI-COR).

## Analog-specific substrate screening for aPKCs

To perform the analog-sensitive screening to detect aPKC substrates, gatekeeper residues (isoleucine in the ATP-binding pocket of aPKCs were point-mutated into glycine to generate analog-specific (AS)-aPKCs. WT-aPKCs and AS-aPKCs were expressed in 293 T. Cells were incubated for 20 min at room temperature in the presence of 25 mM Tris·HCl (pH 7.5), 5 mM MgCl2, 0.5 mM EGTA, 1 mM DTT, 400 μM 6-Bn-ATPγS, 30 μg/mL digitonin, 5 mM GTP and 1X protease and phosphatase inhibitor cocktail. After adding 20 mM EDTA, cells were sonicated and clarified by centrifugation. PNBM was added to a final concentration of 2.5 mM, and the samples were then incubated for 1 h at room temperature. Specific thiophosphorylation sites were detected by LC-MS/MS.

## In vitro phosphorylation assay

293 T cells were transfected with 3 μg of GFP-tagged SCAP WT or triple mutant (AAA) in p60 format. Cells were lysed with IP lysis buffer (100 mM NaCl, 25 mM Tris, 1% Triton X-, 10% glycerol, with phosphatase and protease inhibitors) 48 h after transfection and GFP-SCAP was immunoprecipitated with 25 μL of 50% slurry of recombinant protein G-Sepharose 4B (Invitrogen). Immunoprecipitates were washed and incubated at 30 °C for 60 min in kinase-assay buffer containing 175 mM Tris-HCl (pH 7.5), 50 mM MgCl2, 0.5 mM CaCl2, 2.5 mM EGTA, 1 mM DTT, and 100 μM ATPγS (Biolog) in the presence of recombinant PKCλ/ι (Invitrogen). Detection of substrate phosphorylation was performed using ATP analog-based phosphorylation detection. After the phosphorylation reaction, PNBM (Abcam) and EDTA were added to a final concentration of 2.5 mM and 20 mM, respectively, and incubated for 1 h at room temperature. Immunoblotting detection was performed with anti-thiophosphate ester antibody (Abcam).

## RNA Extraction, qPCR analysis, and 3' RNAseq

Total RNA was extracted by TRIZOL reagent (Invitrogen), followed by purification with the Quick-RNA MiniPrep kit (Zymo Research). A Nanodrop 1000 spectrophotometer (Thermo Scientific) was used for RNA quantification. Complementary DNA (cDNA) was prepared using random primers and MultiScribe Reverse Transcriptase (Applied Biosystems). Real-time qPCR was performed by amplifying 500 ng of the cDNA with the CFX96 Real-Time PCR Detection System and SYBR Green Master Mix (BioRad) with the following parameters: 95 °C for 30 s, 58 °C for 30 s, and 72 °C for 30 s (40 cycles total). Gene expression values were normalized to the 18 S RNA. The sequence for the primers used in qPCR is described in Supplementary Table 1. For 3' RNAseq, 200 ng of total RNA prepared by the Quick-RNA MiniPrep kit (Zymo Research) was used with the QuantSeq 3' mRNA-Seq Library Prep Kit FWD for Illumina from Lexogen, and optional UMIs (Vienna, Austria). Barcoded libraries were pooled, and single end sequenced (1 × 75) on the Illumina NextSeq 500 using the High output V2.5 kit (Illumina Inc., San Diego CA).

## Analysis of single-cell RNA sequencing

After performing the first quality check using FastQC v0.11.7, raw sequence reads were processed aligned to the mouse mm10 assembly reference genome (University of California Santa Cruz, UCSC), and the unique molecular identifier (UMI) counts were summarized using Cell Ranger version 3.0 software suite (10X Genomics, https://support.10xgenomics.com/single-cell-gene-expression/software/downloads/latest). Raw unfiltered UMI count matrices were imported into R version 4.0.3. Barcodes with less than 200 (empty wells) or more than 4000 total UMI counts (doublets), cells expressing more than 12% mitochondrial genes, and cells with fewer than 250 genes were removed using the Seurat v 3.0 R package[82]. Each sample was normalized with SCTransform, regressing out the following variables: total number of UMIs per cell and percentage of mitochondrial UMIs. The top principal components were computed and identified using the ElbowPlot function. The RunHarmony() function from Harmony software v0.1.1, RunUMAP(), FindNeighbors(), and FindClusters() functions were run to correct batch effect among samples and to compute the clustering after regressing the percentage of mitochondrial features out. Cluster gene markers were identified using the FindAllMarkers() function and the Wilcoxon Rank Sum test. Intestinal epithelial cells, identified as Epcam-positive cells, were annotated based on canonical features from the literature. AddModuleScore function in Seurat with default parameters was used to compute the scoring for the indicated signatures. Double-positive, single-positive, or double-negative cell expression was identified for metaplasia and cholesterol signatures with the WhichCells() function. Differential gene expression analysis between clusters was run with the FindMarker() function in Seurat. We used a fast pre-ranked gene set enrichment analysis named fgsea (v1.18.0, R package) with default parameters to perform functional enrichment analysis for defined gene signatures of cluster-specific differentially expressed genes. Functionally enriched

biological states or processes were selected by expression of Gene sets with adjusted *P* value < 0.05. Single-cell RNA-seq data of CRC patients were obtained from GSE166555 or GSE132465. The visualization of the indicated signature scores was performed with Dot Plot or Violin Plot. Statistical analysis (T-test) was calculated using the ggplot2 package in R software in all Violin Plots.

## ATAC-seq library preparation, sequencing and analysis
ATAC-seq libraries were prepared from nuclei pellets using the Nextera DNA Library Prep Kit (Illumina), followed by purification of the transposase-associated DNA with the MinElute PCR purification kit (QIAGEN). The size of the total amplified DNA was limited to 800 bp with SPRI beads. For library quality control, QuBit and TapeStation were used for quatification and size determination, respectively. Barcoded libraries were pooled and paired end sequenced (sX75) on the Illumina NextSeq 500 using the High output V2.5 kit (Illumina Inc., San Diego, CA). FASTQ files were aligned to UCSC mm10 with Bowtie2 (bowtie2 −very-sensitive -x mm10 -1 FILE_merged_R1.fastq −2 FILE_merged_R2.fastq -X 1000 -p 12 | samtools view -u - | samtools sort - > FILE.bam). Peak calling was done with MACS2 with a threshold of q < 0.05 and annotated with ChipSeeker R package. For motif enrichment analysis in the differential peaks between WT and *Prkcz*$^{f/f}$;*Villin-Cre*, *Prkci*$^{f/f}$;*Villin-Cre*, or *Prkci*$^{f/f}$*Prkcz*$^{f/f}$;*Villin-Cre*, p values were calculated using findMotifsGenome.pl with -size given,-len 6,8,10,12 and -mask program of HOMER v4.10.3[83] within 1−2 Kb from the TSS.

## RNA-seq analysis
RNA-seq data for *Prkci*$^{f/f}$*Prkcz*$^{f/f}$;*Villin-Cre* tumors and WT mice, IECs of each mouse genotype, and MTOs of each genotype were described previously (GSE109289, GSE207776)[14,15]. Raw gene expression datasets (GSE79462, GSE76987, and GSE4045) were accessed through GEO website (https://www.ncbi.nlm.nih.gov/geo/).

GenePattern(https://genepattern.broadinstitute.org/gp/pages/index.jsf) was used to collapse gene matrix files (CollapseDataset module) or to assess the statistical significance of differential gene expression (DESeq2). Genes were sorted by log$_2$ FC > 0.5 and adj > 0.05. Gene Set Enrichment Analysis (GSEA) was performed using GSEA 4.0 software (http://www.broadinstitute.org/gsea/index.jsp) with 1000 gene-set permutations using the gene-ranking metric t-test. Gene Set Variation Analysis (GSVA) was performed with GSVA R package[84]. Expression data and clinical data for The Cancer Genome Atlas Colorectal Adenocarcinoma (Tumor Samples with mRNA data (RNA Seq V2 RSEM), 592 tumor samples from 592 patients) (TCGA-COADREAD) were accessed through cBioportal (http://www.cbioportal.org). In the TCGA dataset, CRC patients were categorized into two groups based on the primary tumor location (proximal: cecum, ascending colon, hepatic flexure; and distal: splenic flexure, descending colon, sigmoid colon, rectosigmoid junction, rectum). Expression levels of aPKCs in CRC patients were calculated as mean values of *PRKCI* and *PRKCZ* mRNA levels. *PRKCI* and *TFMC* scores for each patient were calculated by GSVA and used to categorize patients into high and low expression with quartiles as cutoffs.

## Identification of PKCλ/ι-binding partners by BioID2-based screening
A BioID2-based screening was performed in 293 T cells, stably expressing myc-BioID2 or myc-BioID2-PKCλ/ι as bait. Cell pellets were collected after incubation of 50 μM biotin for 48 h by sonication in lysis buffer containing 8 M urea and 50 mM ammonium bicarbonate. For affinity purification of biotinylated proteins, a total of 700 mg of protein extract per sample was used. Cysteine disulfide bonds were reduced by incubation at 30 °C for 60 min in 5 mM tris(2-carboxyethyl) phosphine (TCEP), followed by cysteine alkylation by incubation in the dark at room temperature for 30 min with 15 mM iodoacetamide (IAA). A Bravo AssayMap platform (Agilent) with AssayMap streptavidin

cartridges (Agilent) was used for affinity purification, with priming of cartridges with 50 mM ammonium bicarbonate, proteins loaded onto the streptavidin cartridge and washed with 8 M urea, 50 mM ammonium bicarbonate. Cartridges were treated with Rapid digestion buffer (Promega, Rapid digestion buffer kit) and subjected to mass spec grade Trypsin/Lys-C Rapid digestion enzyme (Promega) at 70 °C for 2 h. Digested peptides were analyzed on a Thermo Fisher Orbitrap Lumos mass spectrometer with an Easy nLC 1200 ultra-high pressure liquid chromatography system. MaxQuant software package and Andromeda search engine (version 1.5.5.1) were used to analyze raw mass spectrometry data. MaxQuant default settings were used except for variable modifications for N-terminal protein acetylation, methionine oxidation, and lysine acetylation, and a static modification for carbamidomethyl cysteine. Peptide and protein identifications were performed against Uniprot human protein database (version August 13, 2015) and filtered to a 1% false discovery rate (FDR). The MSstats software package was used for statistical analysis.

## Lipid analysis
Total lipids were extracted by using Lipid Extraction Kit following the manufacturer's protocols. Tissue lipid concentrations were determined by spectrophotometry (Wako) and normalized to tissue weights. Mouse serum was prepared from retro-orbitally collected whole blood and serum cholesterol levels were measured by spectrophotometry. Cellular cholesterol contents were measured by Amplex Red Cholesterol Assay Kit (Invitrogen) following the manufacturer's protocols and normalized to total protein levels.

## In vitro isotopic tracing
For in vitro de novo lipogenesis assays, 293 T sgC and sg*PRKCI* cells were cultured for 48 h in growth medium with 10% FBS, and then the medium was changed to a growth medium containing 25 mM glucose (25% [U-$^{13}$C$_6$]glucose and 75% $^{12}$C glucose) and 4 mM glutamine with 1% FBS or 1% LPDS. Cells were cultured in tracer media for 24 h. All media was adjusted to pH = 6.8. The fraction of newly synthesized palmitate from [$^{13}$C]glucose was determined via isotopomer spectral analysis (ISA) using INCA. Significance was considered as non-overlapping confidence intervals. Labeling on cholesterol is depicted as $^{13}$C mole percent enrichment (MPE) from [$^{13}$C]glucose relative to the control condition.

## Gas chromatography-mass spectrometry (GC/MS)
Metabolites from tissues and cells were extracted using a modified Bligh and Dyer method. Tissues (10 mg) were homogenized using a Retsch mill for 5 min at 30 I/S. Intracellular metabolites were extracted with 0.25 mL −20 °C methanol, 0.1 mL 4 °C cold water, and 0.25 mL −20 °C chloroform. The extracts were vortexed for 10 min at 4 °C and centrifuged at 16,000 × *g* for 5 min at 4 °C. The dried lower organic phase was derivatized to form fatty acid methyl esters (FAMES) using 500 μL 2% H$_2$SO$_4$ in MeOH and incubation at 50 °C for 2 h. FAMES were extracted via addition of 100 μL saturated salt solution and 500 μL hexane. FAMES were analyzed using a Select FAME column (100 m x 0.25 mm i.d.) installed in an Agilent 7890 A gas chromatograph (GC) interfaced with an Agilent 5975 C mass spectrometer (MS). Helium was used as a carrier gas and the GC oven was held at 80 °C, increased by 20 °C/min to 170 °C, increased by 1 °C/min to 204 °C, then 20 °C/min to 250 °C and hold for 10 min. After measurement, samples were dried under air and derivatized with 15 μL N-methyl-N-(trimethylsilyl)trifluoroacetamid (MSTFA) and 15 pyridine for 25 min at 45 °C using a Gerstel MPS robotic system. Derivatives were analyzed by GC-MS using a DB-35MS column (30 m x 0.25 mm i.d.) installed in an Agilent 7890B GC interfaced with an Agilent 5977 A MS operating under electron impact ionization at 70 eV. The MS source was held at 230 °C and the quadrupole at 150 °C and helium was used as carrier gas. The GC oven was held at

150 °C for 1 min, then to 260 °C at 20 °C/min and held for 3 min, increased to 280 °C at 10 °C/min and held for 15 min.

## Plasma $^2H_2O$ enrichment analysis and de novo lipogenesis calculations

For measurement of de novo lipogenesis in vivo, 12-week-old $Prkci^{f/f}Prkcz^{f/f}$;Villin-Cre mice or WT mice were injected with 27 μL/g body weight of 0.9% NaCl in Deuterium water ($^2H_2O$), and their drinking water was replaced with 8% enriched $D_2O$. Mice were sacrificed 16 h after the injection. The small intestine, tumors, and plasma were collected and snap-frozen. The $^2H$ labelling of water from samples or standards was determined by deuterium acetone exchange. A 5-μL sample or standard was reacted with 4 μL of 10 M NaOH and 4 μL of a 5% (vol:vol) solution of acetone in acetonitrile for 24 h. Acetone was extracted by addition of 600 μL chloroform and 0.5 g $Na_2SO_4$, followed by vigorous mixing. Eighty microlitres of the chloroform was then transferred to a glass gas chromatography (GC)–MS vial. Acetone was measured using an Agilent DB-35MS column (30 m × 0.25 mm internal diameter (i.d.) × 0.25 μm, Agilent J&W Scientific) installed mass spectrometer with the following temperature program: 60 °C initial, increase by 20 °C/min to 100 °C, increase by 50 °C/min to 220 °C and hold for 1 min. The split ratio was 40:1 with a helium flow of 1 mL/min. Acetone eluted at approximately 1.5 min. The mass spectrometer was operated in the electron impact mode (70 eV). The mass ions 58, 59 and 60 were integrated, and the percentage $m_1$ and $m_2$ ($m/z$ 59 and 60) was calculated. Known standards were used to generate a standard curve, and plasma percent enrichment was determined from this. The fraction of newly synthesized lipids (FNS) is determined by the following equation:

$$FNS = ME/(n \times p) \quad (1)$$

Where ME is the average number of deuterium atoms incorporated per molecule ($ME = 1 \times m_1 + 2 \times m_2 + 3 \times m_3 \ldots$), p is the deuterium enrichment in water and n is the maximum number of hydrogen atoms from water incorporated per molecule. N was calculated using the equation: $m_2/m_1 = (N-1)/2 \times p/q$ where q is the fraction of hydrogen atoms and $p + q = 1$. The molar amount of newly synthesized fatty acids was determined by the following equation:

$$MNS = FNX \times total\ lipid\ amount(nmoles/mg\ tissue) \quad (2)$$

## Statistics and reproducibility

All statistical tests were justified for every figure. All samples represent biological replicates. Data are presented as the mean ± SEM. Statistical analysis was performed using GraphPad Prism 9 or R software environment (http://www.r-project.org/). Significant differences between groups were determined using a Student's t-test (two-tailed) when the data met the normal distribution tested by D'Agostino test. Otherwise, a Mann-Whitney U-test was used. Differences between more than 3 groups were determined using one-way ANOVA test (parametric) or Brown-Forsythe and Welch ANOVA tests (nonparametric) followed by Dunnett post hoc test. If the data did not meet this test, a Mann-Whitney test was used. Differences in Kaplan Meier plots were analyzed by the log-rank test. Logistic regression analysis was employed to estimate multivariate odds ratio and 95% confidence interval (CI). Extra sum-of-squares F test was used to compare IC50 between genotypes. Chi-square test was used to compare the contingency. Values of $p < 0.05$ were considered significantly different. No data were excluded from the analyzes. No statistical method was used to predetermine the sample size. Investigators were not blinded to group allocation at the time of data collection and analysis. For each experiment, at least three replicates were used. All attempts at replication generated reproducible results supporting the overall conclusion.

## Reporting summary

Further information on research design is available in the Nature Portfolio Reporting Summary linked to this article.

## Data availability

The RNA-seq, scRNA-seq, and ATAC-seq data generated in this study have been deposited in the GEO database under accession codes GSE239917, GSE236848 and GSE236850 (Superseries, [https://www.ncbi.nlm.nih.gov/geo/query/acc.cgi?acc=GSE236851]). This paper reports data derived from published and publicly available datasets GEO: GSE109289[14], GSE207780[15], GSE132465[85], GSE166555[86], GSE79462[87], GSE76987[88], GSE4045[89], GSE180223[46]. Data for TCGA-COREAD was accessed through cBioportal [https://www.cbioportal.org]. The raw data generated in this study are provided in the Source Data file and deposited in Mendeley Data: https://doi.org/10.17632/3vtwfjh2sb.1. [https://data.mendeley.com/datasets/3vtwfjh2sb/1] Source data are provided with this paper.

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

## Acknowledgements

Research was supported by grants by the National Cancer Institute of the National Institutes of Health under awards numbers: R01CA265892, R01CA250025, and R01CA275846 to J.M.; R01CA246765 to M.T.D.-M; R50CA265332 to A.D; R50CA252146 to T.C.; R01CA234245 to C.M.M. Project support for this research was provided in part by the European Union Horizon's 2020 Grant (REVERT project GA848098), and the Center for Translational Pathology in the Department of Pathology and Laboratory Medi-cine, Weill Cornell Medicine. Y.M. was supported by the Japan Society for the Promotion of Science Cross-border Postdoctoral Fellowship. We thank Tavonna Bryant and Tararin Nikomborirak, the personnel of the Optical Microscopy Core at Weill Cornell Medi-cine, and Eric Rosiek of the Molecular Cytology Core at Memorial Sloan Kettering Cancer Center for technical assistance. J.M. and M.T.D.-M. are Homer T. Hirst III Professors of Oncology in Pathology.

## Author contributions

Conceptualization, J.M., M.T.D-M., Y.M., A.M-O, C.M.M., and T.F.O.; Methodology, J.M., M.T.D-M., T.F.O., C.M.M., and Y.M.; Investiga-tion, Y.M., J.F.L., A.M-O., A.D., T.C-D., H.K., X.Z., Q.H., Y.N., N.N., T.C., G.A., M.R-M., M.R-C., and H.K.; Resources, H.K., M.Y., K.M., A.A-G., D.T-M., J.G-S., P.C-Z., and G.I.; Writing-Original Draft, J.M., M.T.D.-M., Y.M., A.M-O., N.N., and Y.N.; Writing-Review & Editing, all authors; Supervision, J.M., and M.T.D-M., and Funding Acquisition, J.M.; and M.T.D-M.

## Competing interests

The authors declare no competing interests.

## Additional information

[1]Department of Pathology and Laboratory Medicine and Sandra and Edward Meyer Cancer Center, Weill Cornell Medicine, New York, NY 10065, USA. [2]Department of Gastroenterology and Hepatology, Kyoto University Graduate School of Medicine, Kyoto, Japan. [3]Division of Gastroenterology, Department of Internal Medicine, Kobe University Graduate School of Medicine, Kobe, Japan. [4]Department of Endocrinology and Metabolism, Graduate School of Medical Science, Kyoto Prefectural University of Medicine, Kyoto, Japan. [5]Molecular and Cell Biology Laboratory, Salk Institute for Biological Studies, La Jolla, CA 92037, USA. [6]Department of Bioinformatics and Biochemistry, Braunschweig Integrated Centre of Systems Biology (BRICS), Technische Universität Braunschweig, Braunschweig 38106, Germany. [7]Cell and Molecular Biology of Cancer Program, Sanford Burnham Prebys, La Jolla, CA 92037, USA. [8]School of Biological Sciences, Department of Molecular Biology, University of California San Diego, San Diego, CA, USA. [9]Department of Gastroenterological Surgery, Osaka Metropolitan University Graduate School of Medicine, 1-4-3 Asahimachi, Abeno-ku, Osaka city 545-8585, Japan. [10]Department of Histology and Pathology, Faculty of Life Sciences, Universidad Católica de Murcia (UCAM), 30107 Murcia, Spain. [11]Department of Pathology, Santa Lucía General University Hospital (HGUSL), Calle Mezquita sn, 30202 Cartagena, Spain. [12]Department of Clinical Analysis, Santa Lucía General University Hospital (HGUSL), Calle Mezquita sn, 30202 Cartagena, Spain. [13]Division of Endocrinology, Diabetes and Metabolism, Johns Hopkins University School of Medicine, Institute for Fundamental Biomedical Research, Johns Hopkins All Children's Hospital, St, Petersburg, FL, USA. ✉e-mail: mtd4001@med.cornell.edu; jom4010@med.cornell.edu

