## [Peer Review File · Nature Communications]

REVIEWER COMMENTS

Reviewer #1 (Remarks to the Author): (please find an improved version of Rev#1's report as attachment since symbols are not correctly displayed here)

Muta et al. have performed a rigorous study with high-quality data to understand the metabolic alterations associated with aggressive mesenchymal colorectal cancer, classified as serrated tumors. In their model system, loss of both atypical protein kinases (aPKCs; Prkci, Prkcz) in the intestinal epithelium has been shown to be a non-oncogenic driver of this disease. Evaluating mechanism through a metabolic lens they show that the cholesterol biosynthesis pathway (but not fatty acid synthesis) is deregulated in this setting. They distinguish that loss of Prkci phenocopies the loss of both aPKCs. They go on to show the mechanism controlling SREBP2 nuclear accumulation and deregulation of the mevalonate pathway is due to the phosphorylation of the SREBP2 chaperone SCAP. SCAP phosphorylation by PKC ζ reduces the stability of SCAP in a TRC8 E3 ligase-dependent manner and thereby inhibits SREBP2 nuclear translocation. This PKC ζ -SCAP-TRC8-SREBP2 axis is novel. Deregulated SREBP2 is associated with human disease progression, which is consistent with their model. Moreover, inhibition of the mevalonate pathway and its restorative feedback response through treatment with atorvastatin and dipyridamole, respectively, inhibits tumor growth. Thus, low PKC ζ serves as a potential biomarker of mevalonate pathway deregulation, identifying this pathway as a tumor vulnerability that is immediately actionable. A very impressive and complete study. For the most part the writing is clear, however some areas need improvement and a more thorough discussion regarding data interpretation is suggested. These suggestions are made to further strengthen an already strong manuscript.

1. The term 'non-oncogenic' driver is confusing. Loss of aPKCs, and PKC ζ in particular, drives the disease potentially through the deregulation of SREBP2 and the mevalonate pathway. Thus, the term 'non-oncogenic' doesn't seem to fit in the context of this manuscript. This is unclear; either justify or remove.
2. To ensure a well-rounded discussion, the authors may wish to include a comment that the tumor cell of origin may inherently harbor features of high-ANXA10/low-PKC ζ expression and elevation of the mevalonate pathway, which may rationalize why these are characteristics of this disease, as opposed to these observed expression levels being molecular changes associated with disease progression.
3. The loss of PKC ζ appears to increase the rate of tumor cell proliferation, which is consistent with MYC GSEA signatures also being a feature of these tumors. Thus, the mechanism of mevalonate pathway deregulation may also occur in these cancers in response to additional mechanisms that involves multiple pathways. For example, MYC may be contributing to the deregulation of the cholesterol pathway. In human disease, MYC may repress aPKC expression. Including this caveat regarding additional mechanisms in the discussion would provide a more well-rounded interpretation to the data that doesn't diminish the contribution of this manuscript, just puts it into context.

4. Line 217-223: As growth rate of the four genotypes involving the aPKC KO's is not the same, the data interpretation of the scRNA-seq and ATAC-seq data may reflect proliferative index, as opposed to mechanism. The relative growth rate needs to be shown and this caveat should be included as part of data interpretation. This is a cause vs consequence issue that does not need to be resolved in this manuscript, simply explained as an underlying proliferative phenotype that may be contributing to the results.

5. Line 329: 'more efficiently' is not accurate. The authors can't say 'more efficient' as steady state expression is shown and clearly the stabilized SCAPAAA mutant is more highly expressed, which could explain the elevated nuclear SREBP2. To claim elevated efficiency, kinetic experiments to compare rate of translocation would be required. As this is beyond the scope of this manuscript, the authors should simply rephrase their interpretation of this data in the text or remove it for future studies in which rate may be directly addressed.

6. Lines 332-334: did the authors validate kinase activity on the substrate at those sites in vitro? If not, then this statement needs to be toned down as other kinases may also regulate this signaling axis. The AAA mutant of SCAP shows that phosphorylation regulates SCAP stability/activity, which may be regulated by PKC ζ and/or other kinases.

7. Authors should provide overview of what is known regarding SCAP phosphorylation and turnover so the reader can appreciate how their work fits into what is known and what is novel.

Additional suggestions:

1. Figure 1m: goblet and tumor are difficult to distinguish as the color keys are too similar; perhaps use a more distinctive color scheme.

2. Figure 2c: what is the difference between 'min/max'? This needs to be shown or included in the text. Is it 2-fold or 10-fold? How robust is the difference?

3. How was intracellular cholesterol levels measured (line 200)?

4. Figure 5c: it is hard to see the data and difficult to discern significance; increased clarity here would be appreciated.

5. Line 341, Fig 6c-e; was this performed in LPDS? These details are not mentioned in the legend.

6. Line 544: text appears to be copy/pasted from another manuscript as it discusses the introduction of anti-PD-L1 antibody, which is not shown here.

Reviewer #2 (Remarks to the Author):

This manuscript by the Moscat/Diaz-Meco group introduces a novel paradigm for serrated intestinal tumors, a subset of tumors that originate mostly from sessile serrated lesions, and which are much less understood than those originated from conventional adenomas. Previous work from this group revealed loss of the aPKCs in human serrated tumors, leading to the development of mouse model with loss of both atypical PKCs (PKCi and PKCz) in the intestine. These mice show spontaneous development of serrated lesions preferentially in the proximal location of the colon, which eventually progress to invasive adenocarcinomas, as well as a very strongly reactive and immunosuppressive stroma. The present study identified a major therapeutic vulnerability for this type of tumors that involves dysregulated cholesterol biosynthesis, rendering the serrated tumors addicted to this metabolic pathway. The main aPKC driving this effect is PKCi (specifically the loss of this kinase). Interestingly, aPKC loss induces the activation of SREBP2, a master transcription factor for the biosynthesis of cholesterol, leading to its nuclear accumulation. Mechanistically, the authors carried out a thorough analysis that led to the identification of the sterol sensing chaperone SCAP as a key PKCi substrate that becomes degraded and ubiquitinated as a consequence of phosphorylation. Finally, mice experiments clearly demonstrate that inhibition of cholesterol biosynthesis by a statin (in combination with dipyrindamole) has antitumor activity, supporting the concept that up-regulation of cholesterol biosynthesis as a consequence of aPKC loss is a major contributor to the tumorigenic phenotype.

Overall, this is a very impressive and comprehensive study, highly mechanistic, and with major therapeutic implications. The amount of data presented is massive, with well controlled experiments and careful quantitative analysis, which makes data quite convincing.

I have only very minors questions/issues:

1. Is there any mechanistic explanation for the differences between PKCz and PKCi, considering their potential overlapping in substrate specificity? Is SCAP a substrate for PKCz?
2. Can the authors comment on the concentration of statin used in this study? Are the plasma concentrations of atorvastatin within the normal range in patients or are mice receiving an unusually high concentration beyond the normal concentration that hypercholesterolemic patients receive?
3. I believe that the correlation between ANXA10/SREBP is overinterpreted. It may be statistically significant, but R2 is low.
4. Is there any epidemiological data supporting statins as agents for the prevention of serrated tumors? If available, please include or discuss.
5. Authors should considering sending some of the material to Supplementary Information. Figures are unusually dense, with some of the data being unnecessary for the main figures. In some cases, this leads to very small figures that are hard to read or interpret.

Reviewer #3 (Remarks to the Author):

The manuscript by Muta et al provides a comprehensive analysis of the role of cholesterol biosynthesis in the initiation and maintenance of colorectal tumors developing through the serrated pathway.

The manuscript combines extensive mechanistic work with analysis of patient samples and modelling in mice. The authors identify a pathway that is initiated by the loss (or low expression) of atypical PKC-lambda/iota causing reduced phosphorylation and stabilization of SCAP, leading to enhanced translocation of its binding partner SREBP2 into the nucleus. This leads to upregulation of genes involved in cholesterol biosynthesis, and to high levels of cholesterol. The authors subsequently show that cancer cells with low PKC-lambda/iota are dependent on this pathway for their survival and that it can be targeted as an alternative form of therapy involving statins and dipyridamole, a 'potentiator' of the effect of statins.

This work is highly significant to the field of fundamental, translational, and possibly also clinical research on serrated colorectal cancer. It builds on earlier work on PKC-lambda/iota from the same authors and significantly extends it.

The quality of the work is excellent, and the manuscript is written in clear language. The methods section is sufficiently clear to allow the work to be reproduced. All claims/conclusions are supported by various high-end and diverse experimental approaches yielding convincing results in each Figure. Together, the work provides a solid basis for starting to develop clinical applications based on the identified cholesterol-dependency pathway in serrated CRC, be it treatment or prevention. This is comprehensively discussed in the manuscript

I congratulate the authors on this beautiful strong work and advise publication of the manuscript as it is.

Muta et al. have performed a rigorous study with high-quality data to understand the metabolic alterations associated with aggressive mesenchymal colorectal cancer, classified as serrated tumors. In their model system, loss of both atypical protein kinases (aPKCs; *Prkci*, *Prkcz*) in the intestinal epithelium has been shown to be a non-oncogenic driver of this disease. Evaluating mechanism through a metabolic lens they show that the cholesterol biosynthesis pathway (but not fatty acid synthesis) is deregulated in this setting. They distinguish that loss of *Prkci* phenocopies the loss of both aPKCs. They go on to show the mechanism controlling SREBP2 nuclear accumulation and deregulation of the mevalonate pathway is due to the phosphorylation of the SREBP2 chaperone SCAP. SCAP phosphorylation by PKC λ/ι reduces the stability of SCAP in a TRC8 E3 ligase-dependent manner and thereby inhibits SREBP2 nuclear translocation. This PKC λ/ι -SCAP-TRC8-SREBP2 axis is novel. Deregulated SREBP2 is associated with human disease progression, which is consistent with their model. Moreover, inhibition of the mevalonate pathway and its restorative feedback response through treatment with atorvastatin and dipyridamole, respectively, inhibits tumor growth. Thus, low PKC λ/ι serves as a potential biomarker of mevalonate pathway deregulation, identifying this pathway as a tumor vulnerability that is immediately actionable. A very impressive and complete study. For the most part the writing is clear, however some areas need improvement and a more thorough discussion regarding data interpretation is suggested. These suggestions are made to further strengthen an already strong manuscript.

1. The term 'non-oncogenic' driver is confusing. Loss of aPKCs, and PKC λ/ι in particular, drives the disease potentially through the deregulation of SREBP2 and the mevalonate pathway. Thus, the term 'non-oncogenic' doesn't seem to fit in the context of this manuscript. This is unclear; either justify or remove.
2. To ensure a well-rounded discussion, the authors may wish to include a comment that the tumor cell of origin may inherently harbor features of high-ANXA10/low-PKC λ/ι expression and elevation of the mevalonate pathway, which may rationalize why these are characteristics of this disease, as opposed to these observed expression levels being molecular changes associated with disease progression.
3. The loss of PKC λ/ι appears to increase the rate of tumor cell proliferation, which is consistent with MYC GSEA signatures also being a feature of these tumors. Thus, the mechanism of mevalonate pathway deregulation may also occur in these cancers in response to additional mechanisms that involves multiple pathways. For example, MYC may be contributing to the deregulation of the cholesterol pathway. In human disease, MYC may repress aPKC expression. Including this caveat regarding additional mechanisms in the discussion would provide a more well-rounded interpretation to the data that doesn't diminish the contribution of this manuscript, just puts it into context.
4. Line 217-223: As growth rate of the four genotypes involving the aPKC KO's is not the same, the data interpretation of the scRNA-seq and ATAC-seq data may reflect proliferative index, as opposed to mechanism. The relative growth rate needs to be shown and this caveat should be included as part of data interpretation. This is a cause vs consequence issue that does not need to be resolved in this manuscript, simply

explained as an underlying proliferative phenotype that may be contributing to the results.

5. Line 329: 'more efficiently' is not accurate. The authors can't say 'more efficient' as steady state expression is shown and clearly the stabilized SCAPAAA mutant is more highly expressed, which could explain the elevated nuclear SREBP2. To claim elevated efficiency, kinetic experiments to compare rate of translocation would be required. As this is beyond the scope of this manuscript, the authors should simply rephrase their interpretation of this data in the text or remove it for future studies in which rate may be directly addressed.
6. Lines 332-334: did the authors validate kinase activity on the substrate at those sites in vitro? If not, then this statement needs to be toned down as other kinases may also regulate this signaling axis. The AAA mutant of SCAP shows that phosphorylation regulates SCAP stability/activity, which may be regulated by PKC λ/ι and/or other kinases.
7. Authors should provide overview of what is known regarding SCAP phosphorylation and turnover so the reader can appreciate how their work fits into what is known and what is novel.

Additional suggestions:

1. Figure 1m: goblet and tumor are difficult to distinguish as the color keys are too similar; perhaps use a more distinctive color scheme.
2. Figure 2c: what is the difference between 'min/max'? This needs to be shown or included in the text. Is it 2-fold or 10-fold? How robust is the difference?
3. How was intracellular cholesterol levels measured (line 200)?
4. Figure 5c: it is hard to see the data and difficult to discern significance; increased clarity here would be appreciated.
5. Line 341, Fig 6c-e; was this performed in LPDS? These details are not mentioned in the legend.
6. Line 544: text appears to be copy/pasted from another manuscript as it discusses the introduction of anti-PD-L1 antibody, which is not shown here.

POINT-BY-POINT RESPONSES

We thank the reviewers very much for their very positive comments, their careful review, and their efforts to help us improve our manuscript. We have revised the manuscript to address the reviewers' comments and incorporate their suggestions.

Please note that reviewers' comments are in blue.

Reviewer #1

We appreciate very much the positive and thorough evaluation of our manuscript by this reviewer. We are very happy to see that h/she finds our study “rigorous with high-quality data” and “a very impressive and complete study”.

Specific suggestions:

1. The term ‘non-oncogenic’ driver is confusing. Loss of aPKCs, and PKC λ /i in particular, drives the disease potentially through the deregulation of SREBP2 and the mevalonate pathway. Thus, the term ‘non-oncogenic’ doesn’t seem to fit in the context of this manuscript. This is unclear; either justify or remove.

Following this reviewer’s suggestion, we have deleted the term “non-oncogenic”. The sentence now reads: “Those analyses established aPKC deficiency as a driver of serrated mCRC”.

2. To ensure a well-rounded discussion, the authors may wish to include a comment that the tumor cell of origin may inherently harbor features of high-ANXA10/low-PKC λ /i expression and elevation of the mevalonate pathway, which may rationalize why these are characteristics of this disease, as opposed to these observed expression levels being molecular changes associated with disease progression.

Following this reviewer’s suggestion, we have included the sentence “These results suggest that the tumor cell of origin of serrated tumors might inherently harbor high-ANXA10/low-aPKC expression features and an elevation of the mevalonate pathway” in the discussion.

3. The loss of PKC λ /i appears to increase the rate of tumor cell proliferation, which is consistent with MYC GSEA signatures also being a feature of these tumors. Thus, the mechanism of mevalonate pathway deregulation may also occur in these cancers in response to additional mechanisms that involves multiple pathways. For example, MYC may be contributing to the deregulation of the cholesterol pathway. In human disease, MYC may repress aPKC expression. Including this caveat regarding additional mechanisms in the discussion would provide a more well-rounded interpretation to the data that doesn’t diminish the contribution of this manuscript, just puts it into context.

In response to this comment, we have included the following sentence in the discussion: “However, additional molecular mechanisms beyond the direct phosphorylation of SCAP may also contribute to the dysregulation of the cholesterol pathway in these tumors. Thus, the loss of PKC λ /i upregulates MYC targets in the intestinal epithelium (Extended Data Fig. 2), which have been shown to activate cholesterol biosynthesis transcriptionally”.

4. Line 217-223: As growth rate of the four genotypes involving the aPKC KO’s is not the same, the data interpretation of the scRNA-seq and ATAC-seq data may reflect proliferative index, as opposed to mechanism. The relative growth rate needs to be shown and this caveat should be included as part of data interpretation. This is a cause vs consequence issue that does not need to be resolved in this manuscript, simply explained as an underlying proliferative phenotype that may be contributing to the results.

Our previously published data (Nakanishi et al. Immunity, 2018) already showed the proliferation index in vivo for the four genotypes. We have included these data in Figure 1 below to facilitate its revision. As shown in this Figure, the proliferation of mouse intestinal epithelial cells is higher in DKO than in the single knock-outs. The proliferation in intestinal epithelial cells is also increased in PKC λ /i KO but to a lesser extent than DKO, whereas no changes in proliferation were observed in PKC ζ KO. However, the upregulation of the cholesterol biosynthesis is similarly increased in DKO and PKC λ /i KO as shown by levels of cleaved SREBP2 (Fig. 3k,

now new Fig. 3i), by GSEA cholesterol signature (Fig. 2f, now new Fig. 2c) or by RNAseq of SREBP2 targets (Fig. 3b, now new Extended Data Fig. 3c). Furthermore, in our scRNAseq analysis, terminally differentiated enterocytes in DKO mice also showed higher cholesterol signatures than those in WT mice. Considering that enterocytes are non-proliferative and dominant cell types in the intestinal hierarchy, these results indicate that cholesterol dysregulation results from fundamental metabolic reprogramming driven by aPKC loss rather than a mere reflection of cell population shifts towards cycling cell types and is not secondary to proliferative advantage. In addition, we would like to emphasize that although other additional mechanisms, such as MYC, could also contribute as a transcriptional regulator of the mevalonate pathway, our data demonstrate a direct biochemical link through the phosphorylation of SCAP. We believe that our results with the SCAP mutant of the PKC λ /I-specific phosphorylation sites strongly demonstrate the direct mechanistic control of the cholesterol biosynthesis by aPKC and not as a consequence of changes in proliferation.

Figure 1 Reviewer#1. From Nakanishi et al. Immunity, 2018. Figure S2. Ki67 staining of small intestines of the indicated genotypes.

5. Line 329: 'more efficiently' is not accurate. The authors can't say 'more efficient' as steady state expression is shown and clearly the stabilized SCAP^{AAA} mutant is more highly expressed, which could explain the elevated nuclear SREBP2. To claim elevated efficiency, kinetic experiments to compare rate of translocation would be required. As this is beyond the scope of this manuscript, the authors should simply rephrase their interpretation of this data in the text or remove it for future studies in which rate may be directly addressed.

Following this reviewer's suggestion, we have rephrased the interpretation of the data of Fig. 5p,q (now Fig. 6f,g) to state: "the expression of SCAP^{AAA} resulted in increased processing and nuclear accumulation of SREBP2 as compared to SCAP^{WT}".

6. Lines 332-334: did the authors validate kinase activity on the substrate at those sites in vitro? If not, then this statement needs to be toned down as other kinases may also regulate this signaling axis. The AAA mutant of SCAP shows that phosphorylation regulates SCAP stability/activity, which may be regulated by PKC λ /I and/or other kinases.

Yes, the experiment of Fig. 5g shows the in vitro phosphorylation of FLAG-tagged-SCAP^{AAA} by recombinant PKC λ /I as compared to FLAG-tagged-SCAP^{WT}. FLAG immunoprecipitates were extensively washed and eluted to prepare the substrates for the in vitro phosphorylation reaction. The purity of the substrate preparation was verified by Coomassie blue staining. Furthermore, we did not detect phosphorylation of FLAG-SCAP^{WT} in the condition in which no recombinant PKC λ /I was included in the in vitro phosphorylation reaction (first lane, Fig. 5g), which demonstrates the absence of a potential kinase bound to the substrate during the FLAG-immunoprecipitation. Therefore, we believe this experiment clearly demonstrates that PKC λ /I directly phosphorylates these SCAP sites.

7. Authors should provide overview of what is known regarding SCAP phosphorylation and turnover so the reader can appreciate how their work fits into what is known and what is novel.

We have reviewed the current literature on the regulation of SCAP phosphorylation and degradation. To the best of our knowledge, there is only one previously published report (PMID: 27601673) on the phosphorylation of SCAP. This study showed that the cAMP/PKC pathway phosphorylated SCAP and that this phosphorylation

facilitated the processing of SREBP2, in contrast to PKC λ/ι -mediated SCAP phosphorylation that inhibits the processing and activation of SREBP2 and cholesterol biosynthesis. Regarding SCAP degradation and turnover, there have been two reports identifying two ER-membrane ubiquitin ligases that ubiquitinate SCAP: RNF145 and RNF5 (PMID: 29068315 and 32054686). However, none of these ubiquitin ligases control SCAP degradation. RNF145 triggered ubiquitination of SCAP on lysine residues within a cytoplasmic loop essential for COPII binding, potentially inhibiting its transport to Golgi and subsequent processing of SREBP-2, but the abundance of SCAP protein did not change in response to RNF145 expression (PMID: 29068315). RNF5-induced Lys-29 polyubiquitination led to a conformational change of SCAP, which promoted cholesterol biosynthesis (PMID: 29068315). Previous data also showed that TRC8 formed a complex with SCAP and SREBP2 (PMID: 19706601), but it did not investigate its potential direct ubiquitination or impact on SCAP protein levels. Therefore, our data constitutes a novel finding identifying PKC λ/ι -mediated phosphorylation as a key event for TRC8-induced ubiquitination and degradation of SCAP. We have modified the Discussion (see below) to incorporate these previous reports and to put our findings in the appropriate context:

“SCAP phosphorylation recruits the ER-bound TRC8 that promotes SCAP ubiquitination and degradation. Few prior reports have addressed the regulation of SCAP levels and activity by post-translational modifications, including phosphorylation. SCAP has been shown to be phosphorylated by the cAMP/PKA pathway, which promoted SREBP2 processing⁵⁹, in contrast to PKC λ/ι -phosphorylation that represses its activity. Two ER membrane ubiquitin ligases, RNF145 and RNF5, have been reported to induce SCAP ubiquitination with different outputs in cholesterol biosynthesis but without affecting SCAP abundance or stability^{60,61}.”

Additional suggestions:

1. Figure 1m: goblet and tumor are difficult to distinguish as the color keys are too similar; perhaps use a more distinctive color scheme.

We have changed the color scheme in Fig. 1m for the different cell populations to ensure that each population can be clearly distinguished.

2. Figure 2c: what is the difference between ‘min/max’? This needs to be shown or included in the text. Is it 2-fold or 10-fold? How robust is the difference?

The ‘min/max’ scales have been included for Fig. 2c (now Extended Data Fig. 2f), Fig. 2f (now Fig. 2c) and Fig. 3b (now Extended Data Fig. 3c).

3. How was intracellular cholesterol levels measured (line 200)?

Intracellular cholesterol levels were measured by the Amplex Red Cholesterol Assay Kit and normalized to total protein levels as described at the end of “Lipid analysis” section in the Methods.

4. Figure 5c: it is hard to see the data and difficult to discern significance; increased clarity here would be appreciated.

Following this reviewer’s suggestion, we have reformatted Fig. 5c to increase clarity. Similar modifications have been made to Fig. 3m (now Fig. 3k), Fig. 5s (now Fig. 6i) and Fig. 5u (now Fig. 6k).

5. Line 341, Fig 6c-e; was this performed in LPDS? These details are not mentioned in the legend.

Experiments in Fig 6c-e (now Fig. 7c-e) were all performed with 10% FBS. In Fig. 6 (now Fig. 7), LPDS was only used in Fig. 6a,b (now Fig. 7a,b) as indicated in the Figure. We have now included this information also in the legend.

6. Line 544: text appears to be copy/pasted from another manuscript as it discusses the introduction of anti-PD-L1 antibody, which is not shown here.

We apologize for the mistake. We have corrected the Methods section and removed the “Allograft experiments” paragraph.

Reviewer #2

We thank this reviewer for his/her positive comments on our manuscript. We are very happy to see that h/she finds that “this is a very impressive and comprehensive study, highly mechanistic, and with major therapeutic implications. The amount of data presented is massive, with well controlled experiments and careful quantitative analysis, which makes data quite convincing”.

Minor questions/issues:

1. Is there any mechanistic explanation for the differences between PKC ζ and PKC ι , considering their potential overlapping in substrate specificity? Is SCAP a substrate for PKC ζ ?

We have not tested whether SCAP is a substrate of PKC ζ in vitro. However, based on the prediction of motif analyses for PKC ζ and PKC ι consensus sites, SCAP is predicted to be a potential substrate for both aPKCs. This is not surprising, as rightly pointed out by this reviewer because both kinases have potential overlap in substrate specificity. However, we believe other factors, such as level of expression and localization, might impose specificity that can explain their selectivity. Our rationale and starting point were to determine the in vivo phenotype for cholesterol metabolism in the specific intestinal knock-out of each kinase. With this approach, we found that the loss of PKC ι was sufficient to recapitulate the upregulation of the cholesterol metabolism observed in the double knockout. As shown in Fig. 3h and 3k (now Fig. 3f and 3i), whereas PKC ζ KO did not induce the processing of SREBP2, PKC ι KO did it and at the same level as the DKO. Future studies with phospho-SCAP-specific antibodies for the PKC ι sites will allow the interrogation of the selectivity of SCAP phosphorylation for each kinase in vivo in a tissue-dependent manner.

2. Can the authors comment on the concentration of statin used in this study? Are the plasma concentrations of atorvastatin within the normal range in patients or are mice receiving an unusually high concentration beyond the normal concentration that hypercholesterolemic patients receive?

We used 50 mg/kg/day of atorvastatin for our mouse experiments, which is in line with the concentration used in in vivo mouse models in previous studies (PMID 32887721). Our goal was to maximize the anti-tumor effect of cholesterol biosynthesis inhibition. In clinical practice, atorvastatin dose usually ranges from 10 to 80 mg/day for patients with dyslipidemia. However, the clinical dosage is optimized to lower the cholesterol levels in the serum but not in the tissues. Therefore, a higher dose might be necessary to sufficiently reduce the intra-tumoral cholesterol levels to suppress tumor growth.

3. I believe that the correlation between ANXA10/SREBP is overinterpreted. It may be statistically significant, but R² is low.

In response to this question, we have removed Fig. 3s showing ANXA10/SREBP2 correlation.

4. Is there any epidemiological data supporting statins as agents for the prevention of serrated tumors? If available, please include or discuss.

As mentioned in the introduction, there are no current studies in the literature that address the role of cholesterol metabolism in serrated tumors, and there are no epidemiological data that we are aware of on the effect of statins in CRC with an adequate stratification of the disease subsets, including serrated CRC.

5. Authors should consider sending some of the material to Supplementary Information. Figures are unusually dense, with some of the data being unnecessary for the main figures. In some cases, this leads to very small figures that are hard to read or interpret.

Following this reviewer's suggestion, we have reformatted the Figures. We have sent to Supplementary Information the following figures: Fig. 1p-r (now Extended Data Fig. 1c-e), Fig. 1t-u (now Extended Data Fig. 1i-k), Fig. 2a-c (now Extended Data Fig. 2a,c,f), Fig. 3b,c (now Extended Data Fig. 3c,d), Fig. 3q (now Extended Data Fig. 3g), Fig. 4j-m (now Extended Data Fig. 4a-d). We have split Fig. 5 into two Figures (now Fig. 5 and Fig. 6) for a total of eight main figures that are under the allowed number.

Reviewer #3

We appreciate very much the positive revision of our manuscript by this reviewer. We are very happy to see that this reviewer finds our work “highly significant to the field of fundamental, translational, and possibly also clinical research on serrated colorectal cancer” and that h/she recommends its publication as it is.

REVIEWERS' COMMENTS

Reviewer #2 (Remarks to the Author):

The authors addressed my concerns and those from other reviewer. Very strong paper.

POINT-BY-POINT RESPONSES

We thank you very much, Reviewer #2, for taking the time and effort to review our revised manuscript. We are very happy to see that this reviewer finds that we have addressed all his/her concerns and those of the other reviewers and that s/he thinks this is a very strong paper.